# ZoomTrack: Target-aware Non-uniform Resizing for Efficient Visual Tracking

**Yutong Kou**[1,2], **Jin Gao**[1,2*], **Bing Li**[1,5], **Gang Wang**[4*],
**Weiming Hu**[1,2,3], **Yizheng Wang**[4], **Liang Li**[4*]
[1]State Key Laboratory of Multimodal Artificial Intelligence Systems (MAIS), CASIA
[2]School of Artificial Intelligence, University of Chinese Academy of Sciences
[3]School of Information Science and Technology, ShanghaiTech University
[4]Beijing Institute of Basic Medical Sciences    [5]People AI, Inc
kouyutong2021@ia.ac.cn, {jin.gao,bli,wmhu}@nlpr.ia.ac.cn,
liang.li.brain@aliyun.com, g_wang@foxmail.com, yzwang57@sina.com

## Abstract

Recently, the transformer has enabled the speed-oriented trackers to approach state-of-the-art (SOTA) performance with high-speed thanks to the smaller input size or the lighter feature extraction backbone, though they still substantially lag behind their corresponding performance-oriented versions. In this paper, we demonstrate that it is possible to narrow or even close this gap while achieving high tracking speed based on the smaller input size. To this end, we non-uniformly resize the cropped image to have a smaller input size while the resolution of the area where the target is more likely to appear is higher and vice versa. This enables us to solve the dilemma of attending to a larger visual field while retaining more raw information for the target despite a smaller input size. Our formulation for the non-uniform resizing can be efficiently solved through quadratic programming (QP) and naturally integrated into most of the crop-based local trackers. Comprehensive experiments on five challenging datasets based on two kinds of transformer trackers, *i.e.*, OSTrack and TransT, demonstrate consistent improvements over them. In particular, applying our method to the speed-oriented version of OSTrack even outperforms its performance-oriented counterpart by 0.6% AUC on TNL2K, while running 50% faster and saving over 55% MACs. Codes and models are available at `https://github.com/Kou-99/ZoomTrack`.

## 1  Introduction

In visual tracking, many efforts have been made to improve the discrimination and localization ability of the tracking models, including deeper networks [18, 34], transformer feature fusion [8, 31, 21, 19], joint feature extraction and interaction [33, 7, 9], and so on. However, most of the recent tracking algorithms, including the transformer-based [33, 7, 19, 9], still follow the paradigm proposed by Bertinetto et al. [3], in which a small exemplar image cropped from

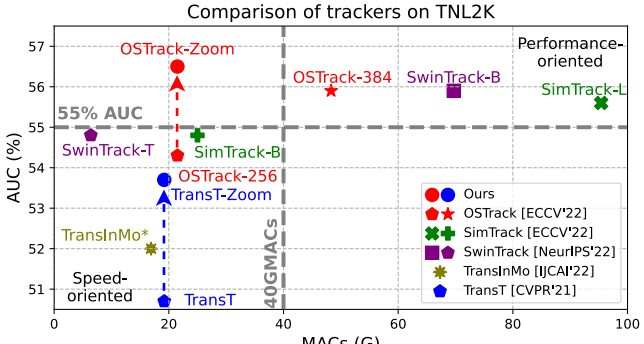

Figure 1: Our method consistently improves the OSTrack and TransT baselines with negligible computation.

---

*Corresponding author

37th Conference on Neural Information Processing Systems (NeurIPS 2023).

the first frame is used to locate the target within a large search image cropped from one of the subsequent frames. The crop size, which determines the visual field size, is derived from a reference bounding box plus a margin for context. Both of the crops are *uniformly* resized to fixed sizes to facilitate the training and testing of the tracker.

Due to the fact that the complexity of the transformers scales quadratically with input size, many transformer trackers [33, 7, 19, 9] propose both speed-oriented versions with the smaller input size or feature extraction backbone and performance-oriented versions with larger ones. Thanks to the enabled larger visual field size or stronger feature representation, the performance-oriented versions always outperform their speed-oriented counterparts in performance, though the tracking speed is severely reduced. For instance, an increased $1.5\times$ input size for OSTrack [33] will cause doubled Multiply–Accumulate Operations (MACs) and nearly halved tracking speed. Thus, it is natural to pose the question: ***Is it possible to narrow or even close this performance gap while achieving high tracking speed based on a smaller input size?***

Inspired by the human vision system (HVS), we propose to non-uniformly resize the attended visual field to have smaller input size for visual tracking in this paper. Human retina receives about 100 MB of visual input per second, while only 1 MB of visual data is able to be sent to the central brain [36]. This is achieved by best utilizing the limited resource of the finite number of cones in HVS. More specifically, the density of cone photoreceptors is exponentially dropped with eccentricity (*i.e.*, the deviation from the center of the retina), causing a more accurate central vision for precise recognition and a less accurate peripheral vision for rough localization [36]. This enables us to solve the dilemma of attending to larger visual field while retaining more raw information for the target area despite the smaller input size.

In our formulation for the non-uniform resizing, the area where the target is more likely to appear is magnified to retain more raw information with high resolution and facilitate precise recognition, whereas the area where the target is less likely to appear is shrunk yet preserved to facilitate rough localization when encountering fast motion or tracking drift. The key to our method is to design an efficient and controllable resizing module based on a small controllable grid. On top of this grid, we define a zoom energy to *explicitly* control the scale of magnification for the important target area, a rigid energy to avoid extreme deformation and a linear constraint to avoid cropping the source image during resizing. The important area is determined by the previous tracking result as a temporal prior. This formulation can be solved efficiently through

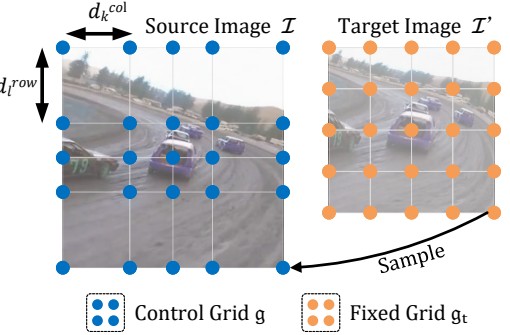

Figure 2: We achieve non-uniform resizing by sampling according to a manipulable control grid $\mathfrak{g}$, which is generated by solving for best $d_k^{col}$ and $d_l^{row}$ that minimize two energies that lead to controllable magnification and less deformation.

quadratic programming (QP), whose solution is used to manipulate the controllable grid. Finally, the ideal non-uniform resizing is achieved by sampling according to the controllable grid (See Fig. 2).

Our method can be easily integrated into plenty of tracking algorithms. We select the popular hybrid CNN-Transformer tracker TransT [8] and one-stream Transformer tracker OSTrack [33] to verify the efficiency and effectiveness of our method. Comprehensive experiments are conducted on five large-scale benchmarks, including GOT-10k [16], LaSOT [11], LaSOT$_{ext}$ [12], TNL2k [30], TrackingNet [23]. We observe consistent improvements over the baselines with negligible computational overhead. In particular, we improve the speed-oriented version of OSTrack to achieve 73.5% AO, 50.5% AUC and 56.5% AUC on GOT-10k, LASOT$_{ext}$ and TNL2k respectively, which are on par with the performance-oriented counterpart while saving over 55% MACs and running 50% faster (see Fig. 1). In other words, the performance gap between speed-oriented and performance-oriented trackers can be narrowed or even closed through our method.

In summary, our contributions are as follows: (**i**) We propose ZoomTrack, an efficient non-uniform resizing method for visual tracking that bridge the gap between speed-oriented and performance-oriented trackers with negligible computation; (**ii**) We formulate the non-uniform resizing as an explicitly controlled magnification of important areas with restriction on extreme deformation,

enabling an HVS-inspired data processing with limited resource; (**iii**) Extensive experiments based on two baseline trackers on multiple benchmarks show that our method achieves consistent performance gains across different network architectures.

## 2   Related Work

**Efficient Visual Tracking.** A lot of real-world applications require tracking algorithms to have high speed. Consequently, many efforts have been made to improve the efficiency of visual tracking. Yan et al. [32] use NAS to search for a lightweight network architecture for visual tracking. Borsuk et al. [6] design a novel way to incorporate temporal information with only a single learnable parameter which achieves higher accuracy at a higher speed. Blatter et al. [5] design an efficient transformer layer and achieves real-time tracking on CPU. Shen et al. [26] propose a distilled tracking framework to learn small and fast trackers from heavy and slow trackers. Some other works [17, 10] explore the quickly advancing field of vision transformer pre-training [15, 29] and architecture designing [13] to improve both tracking accuracy and efficiency. Previous methods focus on designing either a better network [32, 5, 6, 17, 10] or a new training framework [26] to improve the efficiency of trackers, whereas our work focuses on the non-uniform resizing of the input for the sake of efficiency. Our approach is more general and orthogonal to them.

**Non-uniform Resizing for Vision Tasks.** Image resizing is a common image processing operation. Yet, the standard uniform resizing is not always satisfactory in many applications. Avidan and Shamir [1] resize an image by repeatedly carving or inserting seams that are determined by the saliency of image regions. Panozzo et al. [24] use axis-aligned deformation generated from an automatic or human-appointed saliency map to achieve content-aware image resizing. Recasens et al. [25] adaptively resize the image based on a saliency map generated by CNN for image recognition. Zheng et al. [37] learn attention maps to guide the sampling of the image to highlight details for image recognition. Thavamani et al. [27] use the sampling function of [25] to re-sample the input image based on temporal and dataset-level prior for autonomous navigation. Bejnordi et al. [2] learn a localization network to magnify salient regions from the unmediated supervision of [37] for video object detection. Thavamani et al. [28] propose an efficient and differentiable warp inversion that allows for mapping labels to the warped image to enable the end-to-end training of dense prediction tasks like semantic segmentation. Existing methods either use time-consuming feature extractors to generate saliency [1, 24, 25, 37] or use heuristic sampling functions that cannot explicitly control the extent of the magnification, which may cause unsatisfactory magnification or extreme deformation [27, 2, 28]. In contrast, our approach directly generates the saliency map or important area using the temporal prior in tracking and proposes a novel sampling method that can achieve controllable magnification without causing extreme deformation.

## 3   Background and Analysis

### 3.1   Revisiting Resizing in Deep Tracking

Before introducing our method, we first briefly revisit the resizing operation in the deep tracking pipeline. Given the first frame and its target location, visual trackers aim to locate the target in the subsequent frames using a tracking model $\phi_\theta(\mathbf{z}, \mathbf{x})$, where $\mathbf{z}$ is the template patch from the first frame with the target in the center and $\mathbf{x}$ is the search patch from one of the subsequent frames. Both $\mathbf{z}$ and $\mathbf{x}$ are generated following a crop-then-resize manner. We next detail the cropping and resizing operations.

**Image crop generation.** Both the template and search images are cropped based on a reference bounding box $b$, which can be the ground truth annotation for the template image or the previous tracking result for the search image. Denote a reference box as $b = (b^{cx},\ b^{cy},\ b^w,\ b^h)$, then the crop size can be calculated as

$$W = H = \sqrt{(b^w + (f - 1) \times c^w)(b^h + (f - 1) \times c^h)}\,, \tag{1}$$

where $f$ is the context factor controlling the amount of context to be included in the visual field, and $c^w$ and $c^h$ denote unit context amount which is solely related to $b^w$ and $b^h$. Typical choice of unit context amount is $c^w = b^w,\ c^h = b^h$ [33, 31, 9], or $c^w = c^h = (b^w + b^h)/2$ [8, 19]. A higher context factor means a larger visual field and vice versa. Usually, the search image has a larger

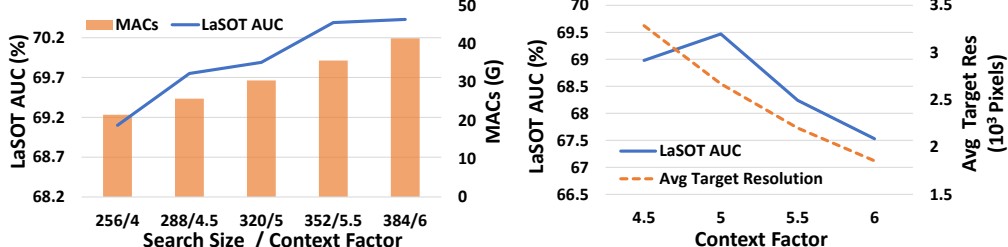

Figure 3: Experiments on LaSOT [11] based on OSTrack [33] show that: (*left*) Thanks to the larger attended visual field, simultaneously increasing search size and context factor leads to consistent performance improvement at the cost of heavy computational burden; (*right*) Increasing the attended visual field by enlarging context factor while fixing the input size to limit the available resources leads to degraded performance due to the decreased target area resolution.

context factor for a larger visual field, and the template image has a small context factor providing minimal necessary context. The image crop $\mathcal{I}$ is obtained by cropping the area centered at $(b^{cx},\ b^{cy})$ with width $W$ and height $H$. Areas outside the image are padded with a constant value. Finally, box $b$ is transformed to a new reference bounding box $r = (r^{cx},\ r^{cy},\ r^{w},\ r^{h})$ on the image crop $\mathcal{I}$.

**Image crop resizing.** To facilitate the batched training and later online tracking, the image crop has to be resized to a fixed size. Given an image crop $\mathcal{I}$ with width $W$ and height $H$, a fix-sized image patch $\mathcal{I}'$ with width $w$ and height $h$ is obtained by resizing $\mathcal{I}$. Specifically, $h \times w$ pixels in $\mathcal{I}'(x',y')$ are sampled from $\mathcal{I}(x,y)$ according to a mapping $\mathcal{T}: \mathbb{R}^2 \to \mathbb{R}^2$, *i.e.*,

$$\mathcal{I}'(x',y') = \mathcal{I}(\mathcal{T}(x',y')) . \tag{2}$$

Note that $\mathcal{T}$ does not necessarily map integer index to integer index. Thus, some pixels in $\mathcal{I}'$ may be sampled from $\mathcal{I}$ according to the non-integer indices. This can be realized by bilinear interpolating from nearby pixels in $\mathcal{I}$. Current methods [33, 9, 19, 7] use *uniform* mapping $\mathcal{T}_{uniform}$ to resize $\mathcal{I}$. $\mathcal{T}_{uniform}$ is defined as $\mathcal{T}_{uniform}(x',y') = \left( \frac{x'}{s^x}, \frac{y'}{s^y} \right)$, where $s^x = w/W$, $s^y = h/H$ are called resizing factors and indicate the scaling of the image. When applying uniform resizing, the resizing factor is the same wherever on the image crop, which means the same amount of amplification or shrinkage is applied to the different areas of the entire image.

### 3.2 Solving the Dilemma Between Visual Field and Target Area Resolution

Although the crop-then-resize paradigm is applied in most of the popular deep trackers, the fixed input search size and context factor vary significantly across different algorithms. Generally, the performance can be improved by using a larger input size at the cost of decreasing tracking speed [19, 33]. To understand the key factors behind this, we conduct a series of experiments on LaSOT [11] based on OSTrack [33]. First, we simultaneously increase search size and context factor, which means the target resolution is roughly fixed and the tracker attends to a larger visual field. As shown in the left of Fig. 3, the AUC on LaSOT increases along with the input search size thanks to the larger attended visual field. Note that increasing the search size ($256 \to 384$) leads to almost doubled computational overhead (21.5G MACs $\to$ 41.5G MACs). Next, we limit the available resources by fixing the input size while increasing the attended visual field size. As shown in the right of Fig. 3, the AUC on LaSOT first increases and then decreases as the average target resolution[2] keeps decreasing. This result demonstrates that the benefit of an enlarged visual field is gradually wiped out by the decrease in target resolution when the computational cost is fixed.

Inspired by the data processing with limited resources in HVS, we believe the above dilemma of attending to a larger visual field while retaining more raw information for the target can be solved by replacing uniform resizing with non-uniform resizing. In visual tracking, the previous tracking result can serve as a strong temporal prior for the current target location. Based on this prior, we can determine the area where the target is likely to appear and magnify it to retain more raw information with high resolution. On the contrary, areas where the target is less likely to appear is shrunk to avoid decreasing the visual field when the input search size is fixed with limited resources. Notably, the

---

[2]The average resolution is calculated by averaging the sizes of the ground truth bounding boxes on the crop-then-resized search patches at frames that do not have full-occlusion or out-of-view attributes.

magnification and shrinkage should not dramatically change the shape and aspect ratio of regions on the image to facilitate the robust learning of the appearance model. Based on the above analysis, we propose some guidelines for our non-uniform resizing, *i.e.*, **G1**: Magnify the area where the target is most likely to appear; **G2**: Avoid extreme deformation; **G3**: Avoid cropping the original image $\mathcal{I}$ so that the original visual field is also retained. We detail how to incorporate these guidelines into our non-uniform resizing in the following section.

## 4 Methods

As discussed in Sec. 3.1, the resizing process from the source image $\mathcal{I}(x, y) \in \mathbb{R}^{H \times W \times 3}$ to the target image $\mathcal{I}'(x', y') \in \mathbb{R}^{h \times w \times 3}$ is a sampling operation controlled by a mapping $\mathcal{T} : (x', y') \to (x, y)$. Our aim is to find the mapping $\mathcal{T}$ that best follows the guidelines (**G1~G3**), given a temporal prior box $r = (r^{cx}, r^{cy}, r^w, r^h)$ that indicates the most possible location of the target. We first restrict the domain of the mapping to a few points and acquire a grid representation of the mapping $\mathcal{T}$. Then, we formulate the guidelines as a QP problem based on the grid point intervals. By solving the QP problem via a standard solver [20], we can efficiently manipulate the controllable grid point intervals and achieve the ideal non-uniform resizing by sampling according to the final grid representation.

### 4.1 Grid Representation for Non-uniform Resizing

Recall the sampling operation in Eq. (2), pixel $(x', y')$ on the target image $\mathcal{I}' \in \mathbb{R}^{h \times w \times 3}$ is sampled from location $(x, y)$ on the source image $\mathcal{I} \in \mathbb{R}^{H \times W \times 3}$. Thus, at least $h \times w$ pairs of $(x', y') \to (x, y)$ are needed for resizing. We use the exact $h \times w$ pairs to represent the mapping $\mathcal{T}$, which can be defined as a grid $\mathcal{G} \in \mathbb{R}^{h \times w \times 2}$, where $\mathcal{G}[y'][x'] = (x, y)$, $x' = 1, ..., w$, $y' = 1, ..., h$, is the location on $\mathcal{I}$ that the target image pixel $\mathcal{I}'(x', y')$ should be sampled from.

**Controllable grid.** The grid $\mathcal{G}$ is an extremely dense grid with $h \times w$ grid points (equivalent to the resolution of the target image $\mathcal{I}'$), which makes it computationally inefficient to be directly generated. To solve this issue, we define a small-sized controllable grid $\mathfrak{g} \in \mathbb{R}^{(m+1) \times (n+1) \times 2}$ with $m + 1$ rows and $n + 1$ columns ($m << h, n << w$) to estimate $\mathcal{G}$, where

$$\mathfrak{g}[j][i] = \mathcal{G}\left[\frac{h}{m}j\right]\left[\frac{w}{n}i\right], \; i = 0, ..., n, \; j = 0, ..., m \tag{3}$$

Intuitively, $\mathfrak{g}[j][i]$ is the sample location on the source image for the target image pixel $(\frac{w}{n}i, \frac{h}{m}j)$ in $\mathcal{I}'$. Once $\mathfrak{g}$ is determined, $\mathcal{G}$ can be easily obtained through bilinear interpolation on $\mathfrak{g}$.

**Axis-alignment constraint.** To avoid extreme deformation (**G3**) and reduce computational overhead, we add an axis-aligned constraint [27] to the grid $\mathfrak{g}$. Specifically, grid points in the same row $j$ have the same y-axis coordinate $y_j$, and grid points in the same column $i$ have the same x-axis coordinate $x_i$. That is to say $\mathfrak{g}[j][i] = (x_i, y_j)$, $i = 0, ..., n$, $j = 0, ..., m$. The reason why we use such an axis-aligned constraint is that we need to keep the axis-alignment of the bounding box (*i.e.*, the four sides of the bounding box are parallel to the image boundary) after resizing, which is proven to be beneficial to the learning of object localization [27].

**Grid representation.** The source image is split into small rectangular patches $p(k, l), k = 1, ...n, l = 1, ..., m$ by grid points. The top-left corner of $p(k, l)$ is $\mathfrak{g}[l - 1][k - 1]$ and its bottom-right corner is $\mathfrak{g}[l][k]$. The width and height of the patch $p(k, l)$ are denoted as the horizontal grid point interval $d_k^{col}$ and the vertical grid point interval $d_l^{row}$. $d_k^{col}$ and $d_l^{row}$ can be calculated as $d_k^{col} = x_k - x_{k-1}$, $d_l^{row} = y_l - y_{l-1}$. The axis-aligned controllable grid $\mathfrak{g}$ can be generated from grid point intervals $d_l^{row}, d_k^{col}$ based on

$$\mathfrak{g}[j][i] = \left(\sum_{k=0}^{i} d_k^{col}, \sum_{l=0}^{j} d_l^{row}\right), \tag{4}$$

where $d_0^{row} = d_0^{col} = 0$ for the sake of simplicity.

**Non-uniform resizing.** According to Eq. (3), the resizing process can be visualized using two grids (see Fig. 2): a manipulatable grid on the source image $\mathfrak{g}$ and a fixed grid on the target image $\mathfrak{g}_t[j][i] = (\frac{w}{n}i, \frac{h}{m}j)$. $\mathfrak{g}$ and $\mathfrak{g}_t$ split the source and target images into rectangular patches $p(k, l)$ and $p_t(k, l)$ respectively. Target image pixel at $\mathfrak{g}_t[j][i]$ is sampled from $\mathfrak{g}[j][i]$ on the source image. This

process can be intuitively understood as resizing $p(k, l)$ to $p_t(k, l)$. When the grid point intervals of $\mathfrak{g}$ are manipulated, the grid point intervals of $\mathfrak{g}_t$ remain unchanged. Thus, by reducing/ increasing the grid intervals of $\mathfrak{g}$, the corresponding region on the target image is amplified/ shrunk. In this way, a non-uniform resizing operation that scales different image areas with different ratios can be obtained by dynamically manipulating the grid point intervals.

## 4.2 QP Formulation Based on Grid Representation

Given a reference bounding box $r = (r^{cx}, r^{cy}, r^w, r^h)$, we aim to dynamically manipulate the grid point intervals to best follow the guidelines (**G1~G3**) proposed in Sec. 3.2.

**Importance score for the target area.** According to **G1**, areas, where the target is most likely to appear, should be magnified. Following this guideline, we first define the importance score for the target area on the source image. An area has high importance if the target is more likely to appear in this area. As $r$ is a strong temporal prior for the current target location, we use a Gaussian function $G(x, y)$ centered at $(r^{cx}, r^{cy})$ as the importance score function, *i.e.*,

$$G(x, y) = \exp\left(-\frac{1}{2}\left(\frac{(x - \mu_x)^2}{\sigma_x^2} + \frac{(y - \mu_y)^2}{\sigma_y^2}\right)\right) , \tag{5}$$

where $\mu_x = r^{cx}$, $\mu_y = r^{cy}$, $\sigma_x = \sqrt{\beta \times r^w}$, $\sigma_y = \sqrt{\beta \times r^h}$. $\beta$ is a hyper-parameter controlling the bandwidth of the Gaussian function. Since we denote the rectangular area enclosed by the grid point intervals $d_l^{row}$ and $d_k^{col}$ as patch $p(k, l)$, its importance score $S_{k,l}$ can be determined by the value of $G(x, y)$ at the center of patch $p(k, l)$, *i.e.*,

$$S_{k,l} = G\left(\left(k + \frac{1}{2}\right) \times \frac{W}{n}, \left(l + \frac{1}{2}\right) \times \frac{H}{m}\right) + \epsilon , \tag{6}$$

where $\epsilon$ is a small constant to prevent extreme deformation and ensure stable computation. Here we use the patch center of a uniformly initialized grid ($d_k^{col} = \frac{W}{n}$ and $d_l^{row} = \frac{H}{m}$). By doing so, $S_{k,l}$ is irrelevant of $d_k^{col}$ and $d_l^{row}$, which is crucial for the QP formulation in the following.

**Zoom energy.** Following **G1**, we design a quadratic energy $E_{zoom}$ to magnify the area where the target is most likely to appear. To achieve controllable magnification, we define a zoom factor $\gamma$ controlling the amount of magnification. If we want to magnify some area by $\gamma$, the distances between sampling grid points should shrink by $\frac{1}{\gamma}$. $E_{zoom}$ forces $d_k^{col}$ and $d_l^{row}$ to be close to the grid point intervals under uniform magnification by $\gamma$, *i.e.*, $\frac{1}{\gamma}\frac{H}{m}$ for $d_l^{row}$ and $\frac{1}{\gamma}\frac{W}{n}$ for $d_k^{col}$. In detail, The zoom energy is defined as

$$E_{zoom} = \sum_{l=1}^{m}\sum_{k=1}^{n} S_{k,l}^2\left(\left(d_l^{row} - \frac{1}{\gamma}\frac{H}{m}\right)^2 + \left(d_k^{col} - \frac{1}{\gamma}\frac{W}{n}\right)^2\right) . \tag{7}$$

**Rigid energy.** To avoid extreme deformation (**G2**), we define a quadratic energy $E_{rigid}$ to restrict the aspect ratio change of patch $p(k, l)$, which has width $d_k^{col}$ and height $d_l^{row}$. When being uniformly resized, $p(k, l)$ should have width $\frac{W}{n}$ and height $\frac{H}{m}$. Thus, the patch $p(k, l)$ is horizontally stretched by $\frac{n}{W}d_k^{col}$ and vertically stretched by $\frac{m}{H}d_l^{row}$. To keep the aspect ratio mostly unchanged for the important area, the horizontal stretch should be similar to the vertical stretch. Therefore, the rigid energy can be defined as

$$E_{rigid} = \sum_{l=1}^{m}\sum_{k=1}^{n} S_{k,l}^2\left(\frac{m}{H}d_l^{row} - \frac{n}{W}d_k^{col}\right)^2 . \tag{8}$$

**Linear constraint to avoid cropping the source image.** According to **G3**, the resizing should not crop the source image so that the original visual field is also retained. That is to say, the grid $\mathfrak{g}$ should completely cover the entire source image. This requirement can be interpreted as a simple linear constraint that forces the sum of vertical grid point intervals $d_l^{row}$ to be $H$ and the sum of horizontal grid point intervals $d_k^{col}$ to be $W$: $\sum_{l=1}^{m} d_l^{row} = H$, $\sum_{k=1}^{n} d_k^{col} = W$.

**QP formulation.** Combining the zoom energy, the rigid energy and the linear constraint, we can formulate the grid manipulation as the following minimization task

$$\underset{d_l^{row},\ d_k^{col}}{\text{minimize}} \quad E = E_{zoom} + \lambda E_{rigid}$$

$$\text{subject to} \quad \sum_{l=1}^{m} d_l^{row} = H, \quad \sum_{k=1}^{n} d_k^{col} = W \tag{9}$$

where $\lambda$ is a hyper-parameter to balance the two energies. This convex objective function can be efficiently solved by a standard QP solver [20] (see Appendix A for more detailed derivations).

## 4.3  Integration with Common Trackers

**Reference box generation.** Our non-uniform resizing method needs a temporal prior reference box $r = (r^{cx},\ r^{cy},\ r^w,\ r^h)$ on the cropped source image $\mathcal{I}$. During testing, $r$ is directly generated from the previous frame tracking result (see Sec. 3.1). During training, we only need to generate $r^w$ and $r^h$ from the ground truth width $g^w$ and height $g^h$ as the center of the temporal prior reference box is fixed at $r^{cx} = W/2$ and $r^{cy} = H/2$. In detail, with probability $q$ we use a smaller jitter $j = j_s$, and with probability $1 - q$ we use a larger jitter $j = j_l$. $r^w$ and $r^h$ are calculated as $r^w = e^{J^w} g^w$, $r^h = e^{J^h} g^h$, where $J^w,\ J^h \sim \mathcal{N}(0, j^2)$.

**Label mapping.** The classification loss of the network $\phi_\theta(\mathbf{z}, \mathbf{x})$ is calculated on the target image $\mathcal{I}'$. Therefore, the classification labels on $\mathcal{I}'$ should be generated from the ground truth bounding box on $\mathcal{I}'$. The common practice is to use normalized ground truth bounding box for the reason that the normalized coordinates remain the same on $\mathcal{I}$ and $\mathcal{I}'$ under uniform resizing. However, this does not hold true for our non-uniform resizing. Therefore, we need to map the ground truth bounding box from $\mathcal{I}$ to $\mathcal{I}'$ via the grid $\mathcal{G}$. Specifically, we map the top-left corner and bottom-right corner of the bounding box as follows. Given a point on the source image $(x, y)$, we first find two grid points on $\mathcal{G}$ that are closest to $(x, y)$. Then, the corresponding point $(x', y')$ can be obtained by bilinear interpolation between the indices of these two points.

**Reverse box mapping.** The bounding box predicted by the network $\phi_\theta(\mathbf{z}, \mathbf{x})$ is on $\mathcal{I}'$ and needed to be mapped back to $\mathcal{I}$ as the final tracking result. We map the top-left corner and bottom-right corner of the predicted bounding box from $\mathcal{I}'$ to $\mathcal{I}$ in a reversed process of label mapping. Specifically, given a point $(x', y')$ on $\mathcal{I}'$, we first find the two grid points of $\mathcal{G}$ whose indices are closest to $(x', y')$, and then the corresponding point $(x, y)$ can be obtained by bilinear interpolation between the values of these points. To minimize the impact of the reverse mapping, we directly calculate the regression loss on $\mathcal{I}$ instead of on $\mathcal{I}'$.

## 5  Experiments

### 5.1  Implementation Details

We apply our method to both one-stream tracker OSTrack [33] (denoted as OSTrack-Zoom) and CNN-Transformer tracker TransT [8] (denoted as TransT-Zoom). The proposed non-uniform resizing module is applied in both the training and testing stages on the search image. We do not apply the proposed non-uniform resizing on the template image because it degrades performance (see Sec. 5.3). The search patch size ($256 \times 256$) is the same as the baseline methods, whereas the context factor is enlarged from 4 to 5. We use a controllable grid $\mathfrak{g}$ with shape $17 \times 17 (m = n = 16)$ . The importance score map is generated by a Gaussian function with bandwidth $\beta = 64$. We set magnification factor $\gamma = 1.5$ and $\lambda = 1$ in Eq. (9). We use smaller jitter $j_s = 0.1$ with probability $0.8$ and larger jitter $j_l = 0.5$ with probability $0.2$. Notably, we use the same set of parameters for both OSTrack-Zoom and TransT-Zoom to demonstrate the generalization capability of our method. Except for the resizing operation, all the other training and testing protocols follow the corresponding baselines. The trackers are trained on four NVIDIA V100 GPUs. The inference speed and MACs are evaluated on a platform with Intel i7-11700 CPU and NVIDIA V100 GPU.

### 5.2  State-of-the-art Comparison

We compare our method with the state-of-the-art trackers on five challenging large-scale datasets: GOT-10k [16], LaSOT [11], LaSOT$_{ext}$ [12], TNL2K [30] and TrackingNet [23].

Table 1: State-of-the-art Comparison on GOT-10k, LaSOT, LaSOT$_{ext}$, TNL2K and TrackingNet.

| | Tracker | Size | GOT-10k | | | LaSOT | | LaSOT$_{ext}$ | | TNL2K | | TrackingNet | | MACs (G) | FPS |
|---|---|---|---|---|---|---|---|---|---|---|---|---|---|---|---|
| | | | AO | SR$_{0.5}$ | SR$_{0.75}$ | AUC | P | AUC | P | AUC | P | SUC | P | | |
| Baseline &Ours | OSTrack-Zoom | 256 | **73.5** | **83.6** | **70.0** | **70.2** | **76.2** | **50.5** | **57.4** | **56.5** | **57.3** | **83.2** | **82.2** | 21.5 | 100 |
| | OSTrack-256[33] | 256 | 71.0 | 80.4 | 68.2 | 69.1 | 75.2 | 47.4 | 53.3 | 54.3 | - | 83.1 | 82.0 | 21.5 | 119 |
| | TransT-Zoom | 255 | 67.5 | 77.6 | 61.3 | 67.1 | 71.6 | 46.8 | 52.9 | 53.7 | 62.3 | 81.8 | 80.2 | 19.2 | 45 |
| | TransT[8] | 255 | 67.1 | 76.8 | 60.9 | 64.9 | 69.0 | 44.8 | 52.5 | 50.7 | 51.7 | 81.4 | 80.3 | 19.2 | 48 |
| Speed-oriented | SwinTrack-T[19] | 224 | 71.3 | 81.9 | 64.5 | 67.2 | 70.8 | 47.6 | 53.9 | 53.0 | 53.2 | 81.1 | 78.4 | 6.4 | 98 |
| | SimTrack-B[7] | 224 | 68.6 | 78.9 | 62.4 | 69.3 | - | - | - | 54.8 | 53.8 | 82.3 | - | 25.0 | 40 |
| | MixFormer-22k[9] | 320 | 70.7 | 80.0 | 67.8 | 69.2 | 74.7 | - | - | - | - | 83.1 | 81.6 | 23.0 | 25 |
| | ToMP-101[21] | 288 | - | - | - | 68.5 | 73.5 | 45.9 | - | - | - | 81.5 | 78.9 | - | 20 |
| | TransInMo*[14] | 255 | - | - | - | 65.7 | 70.7 | - | - | 52.0 | 52.7 | 81.7 | - | 16.9 | 34 |
| | Stark-ST101[31] | 320 | 68.8 | 78.1 | 64.1 | 67.1 | - | - | - | - | - | 82.0 | - | 28.0 | 32 |
| | AutoMatch[35] | 255 | 65.2 | 76.6 | 54.3 | 58.3 | 59.9 | 37.6 | 43.0 | 47.2 | 43.5 | 76.0 | 72.6 | - | 50 |
| | DiMP[4] | 288 | 61.1 | 71.7 | 49.2 | 56.9 | 56.7 | 39.2 | 45.1 | 44.7 | 43.4 | 74.0 | 68.7 | 5.4 | 40 |
| | SiamRPN++[18] | 255 | 51.7 | 61.6 | 32.5 | 49.6 | 49.1 | 34.0 | 39.6 | 41.3 | 41.2 | 73.3 | 69.4 | 7.8 | 35 |
| Performance -oriented | OSTrack-384[33] | 384 | 73.7 | 83.2 | 70.8 | 71.1 | 77.6 | 50.5 | 57.6 | 55.9 | - | 83.9 | 83.2 | 48.3 | 61 |
| | SwinTrack-B[19] | 384 | 72.4 | 80.5 | 67.8 | 71.3 | 76.5 | 49.1 | 55.6 | 55.9 | 57.1 | 84.0 | 82.8 | 69.7 | 45 |
| | SimTrack-L[7] | 224 | 69.8 | 78.8 | 66.0 | 70.5 | - | - | - | 55.6 | 55.7 | 83.4 | - | 95.4 | - |
| | MixFormer-L[9] | 320 | - | - | - | 70.1 | 76.3 | - | - | - | - | 83.9 | 83.1 | 127.8 | 18 |

**GOT-10k** [16] provides training and test splits with zero-overlapped object classes. Trackers are required to be only trained on GOT-10k training split and then evaluated on the test split to examine their generalization capability. We follow this protocol and report our results in Tab. 1. Our OSTrack-Zoom achieves 73.5% AO which surpasses the baseline OSTrack by 2.5%. And our TransT-Zoom achieves 67.5% AO which surpasses the baseline TransT by 0.4%. Compared with previous speed-oriented trackers, our OSTrack-Zoom improves all of the three metrics in GOT-10k by a large margin, proving that our method has good generalization capability.

**LaSOT** [11] is a large-scale dataset with 280 long-term video sequences. As shown in Tab. 1, our method improves the AUC of OSTrack and TransT by 1.1% and 2.2% respectively. Our method OSTrack-Zoom have the best performance among speed-oriented trackers on LaSOT, indicating that our method is well-suited for complex object motion in long-term visual tracking task.

**LaSOT$_{ext}$** [12] is an extension of LaSOT with 150 additional videos containing objects from 15 novel classes outside of LaSOT. In Tab. 1, our method achieves 3.1% and 2.0% relative gains on AUC metric for OSTrack and TransT baselines respectively. Our method OSTrack-Zoom achieves promising 50.5% AUC and 57.4% Precision, outperforming other methods by a large margin.

**TNL2K** [30] is a newly proposed dataset with new challenging factors like adversarial samples, modality switch, *etc*. As reported in Tab. 1, our method improves OSTrack and TransT by 2.2% and 3.0% on AUC. Our OSTrack-Zoom sets the new state-of-the-art performance on TNL2K with 56.5% AUC while running at 100 fps.

**TrackingNet** [23] is a large-scale short-term tracking benchmark. Our method boosts the performance of OSTrack and TransT by 0.1% and 0.4% SUC. The reason for the relatively small improvements is that, contrary to other datasets, TrackingNet has fewer sequences with scenarios that can benefit from an enlarged visual field, *e.g.*scenarios with drastic movement or significant drift after some period of time. Please refer to Appendix F for a detailed analysis.

**Comparison with performance-oriented trackers.** We additionally compare our method with the performance-oriented versions of the SOTA trackers. On the one hand, on GOT-10k, LaSOT$_{ext}$, TNL2K whose test splits object classes have no or few overlap with training data, our OSTrack-Zoom is on-par with the SOTA tracker OSTrack-384, while consuming only 44.5% MACs of the latter and running 50% faster. On the other hand, on LaSOT and TrackingNet whose test splits have many object classes appearing in training data, our OSTrack-Zoom is slightly inferior to some SOTA trackers. A possible reason is that these SOTA trackers with heavier networks fit the training data better, which is

Table 2: Comparison of accuracy, latency and target size using different resizing methods with OSTrack [33].

| # | Method | Size | TNL2K [30] | | LaSOT [11] | | Target Size ($10^3$) | | Latency (ms) | | FPS |
|---|--------|------|------|------|------|------|------|------|--------|------|-----|
| | | | AUC | P | AUC | P | avg↑ | std↓ | resize | net | |
| ① | Uniform | 256 | 55.5 | 55.4 | 69.5 | 75.2 | 2.7 | **24.8** | **0.04** | **8.40** | **119** |
| ② | Uniform | 320 | 56.1 | 56.8 | 69.9 | 76.2 | 4.2 | 38.8 | 0.04 | 10.92 | 92 |
| ③ | FOVEA | 256 | 54.2 | 54.5 | 67.6 | 73.3 | 4.4 | 40.8 | 3.37 | 8.40 | 85 |
| ④ | Ours | 256 | **56.5** | **57.3** | **70.2** | **76.2** | **4.9** | 25.0 | 1.58 | 8.40 | 100 |

Table 3: Ablation of hyper-parameters on LaSOT [11] with OSTrack [33].

(a) Bandwidth $\beta$

| $\beta$ | 4 | 64 | 256 |
|---------|-----|------|------|
| AUC | 69.1 | **70.2** | 69.6 |

(b) Control grid size

| Size | 8 | 16 | 32 |
|------|-----|------|------|
| AUC | 66.4 | **70.2** | 69.5 |

(c) Zoom factor $\gamma$

| $\gamma$ | 1.25 | 1.5 | 1.75 |
|----------|------|------|------|
| AUC | 69.7 | **70.2** | 69.4 |

(d) Balance factor $\lambda$

| $\lambda$ | 0 | 1 | 2 |
|-----------|------|------|------|
| AUC | 69.8 | **70.2** | 69.6 |

Table 4: Ablation on the effect of applying non-uniform resizing in training and testing stages. Results are reported in AUC (%) metric.

| # | Non-uniform resizing | | OSTrack[33] | | TransT[8] | |
|---|---------|----------|-----------|-----------|-----------|-----------|
| | Testing | Training | LaSOT[11] | TNL2K[30] | LaSOT[11] | TNL2K[30] |
| ① | | | 69.1 | 54.3 | 64.9 | 50.7 |
| ② | ✔ | | 69.1 | 55.9 | 66.3 | 53.6 |
| ③ | ✔ | ✔ | **70.2** | **56.5** | **67.1** | **53.7** |

Table 5: Ablation on the effect of applying non-uniform resizing on the template and search images with OSTrack [33].

| # | Non-uniform Resizing | | LaSOT[11] | |
|---|----------|--------|------|------|
| | Template | Search | AUC | P |
| ① | ✔ | | 64.8 | 71.0 |
| ② | | ✔ | **70.2** | **76.2** |
| ③ | ✔ | ✔ | 69.6 | 75.5 |

beneficial to track objects with classes that have been seen during training. Nevertheless, the gap between speed-oriented and performance-oriented trackers is narrowed using our method.

## 5.3 Ablation Experiment

**Comparison with other resizing methods.** As shown in Tab. 2, we compare our non-uniform resizing method (④) with uniform resizing with small/large input sizes (①/②) and FOVEA [27] non-uniform resizing (③). On both TNL2K and LaSOT, our method outperforms other methods. Notably, while applying FOVEA causes a significantly degraded performance (①*vs.*③), our method can achieve large performance gains (①*vs.*④). To understand the reason behind this, we calculate the average target size (avg) and the standard deviation of the target size (std), which represent the degree of the targets' scale change, on LaSOT. Compared with uniform resizing (①*vs.*④), our method achieves $1.34^2 \times$ magnification on average target size, while the standard deviation almost keeps the same. The magnification is close to the desired $\gamma^2 = 1.5^2$, showing that our method achieves controllable magnification. In contrast to FOVEA (③*vs.*④), which severely reduces the tracking performance, our standard deviation is almost unchanged, demonstrating that our method can avoid extreme deformation that severely changes the scale of the target.

**Computational overhead analysis.** Our resizing module runs purely on CPU. The latency for resizing an image is 1.58 ms on an Intel i7-11700 CPU. Compared with uniform resizing (①*vs.*④), our method introduces an additional 1.54 ms latency during resizing, causing a slightly slower speed. However, our performance is better than the uniform resizing counterparts. Especially when compared with uniform resizing with a larger input size (②*vs.*④), our method reaches higher performance while running faster, demonstrating the superiority of using non-uniform resizing in visual tracking. Our method is also faster than FOVEA resizing while having a better performance. Moreover, the MACS for our resizing an image is 1.28 M MACS ($10^6$), which is negligible compared to the network inference which is usually tens of G MACs ($10^9$).

**Resizing in different stages.** It is worth noting that our method can be directly applied to off-the-shell tracking models, *i.e.*, only applying non-uniform resizing method at testing time. Results on LaSOT [11] and TNL2K [30] are reported in Tab. 4 using the AUC (%) metric. As shown in Tab. 4, directly applying our method to the off-the-shelf trackers only in testing can already improve the tracking accuracy (①*vs.*②). Further applying our resizing in training can also boost the performance, owning to the aligned training and testing protocols (②*vs.*③).

**Resizing template and search images separately.** We also investigate the effects of applying the non-uniform resizing on the template and search images separately. As shown in Tab. 5 either applying non-uniform resizing on the template image alone or on both the template and search images performs worse than applying it only on the search image. We think the reason is that the template

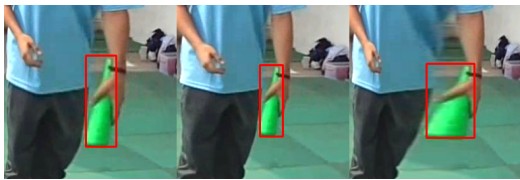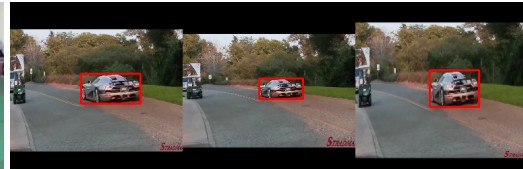

Figure 4: Visualization of different resizing. For each set of images, from left to right: ours, uniform resizing, FOVEA [27] resizing. The targets are marked in red box. (*left*) Our method can magnify the target without drastically changing the appearance of the target. (*right*) Our method can magnify the target without significantly changing the aspect ratio of the target.

image must have a relatively small context factor to contain minimal necessary context, which makes the whole template region important for the tracking network to locate the target. Therefore, the tracker cannot benefit from non-uniform resizing of the template image.

**More ablations. First**, we study the impact of the bandwidth $\beta$ used in the importance function. A smaller $\beta$ means less area is considered to be important and vice versa. As shown in Tab. 3a, the ideal value for $\beta$ is 64. Both smaller bandwidth (4) and larger bandwidth (256) cause reduced performance, showing the importance of this factor. **Second**, we also analyze the impact of the controllable grid. As shown in Tab. 3b, $m = n = 16$ is the best size for this grid. A smaller grid size cause a drastic drop in AUC, showing that too small grid size will cause inaccurate resizing that damages the tracking performance. A larger grid size also degrades performance, for which we believe the reason is that a large grid size will lead to an over-reliance on the important area generation based on the temporal prior, which is however not always correct. **Third**, we further look into the impact of different zoom factors $\gamma$. As shown in Tab. 3c, 1.5 is the best value for $\gamma$. A smaller $\gamma$ will degrade performance due to smaller target resolution. Meanwhile, a larger $\gamma$ is also harmful to the performance as larger magnification will cause trouble when trying to rediscover the lost target. **Finally**, we study the impact of $E_{rigid}$ by changing the balance factor $\lambda$. As shown in Tab. 3d, setting $\lambda = 1$ reaches the best performance. When $\lambda = 0$, the performance drops, demonstrating the effectiveness of the proposed $E_{energy}$. A larger $\lambda = 2$ also degrades the performance, indicating that too much constraint is harmful to the performance.

**Visualization.** We visualize our proposed non-uniform resizing, uniform resizing and FOVEA [27] resizing in Fig. 4. Compared with uniform resizing, our method can improve the resolution of the target to retain more raw information. Compared with FOVEA, our method can better preserve the aspect ratio and the appearance of the target.

## 6   Conclusion

In this paper, we presented a novel non-uniform resizing method for visual tracking, which efficiently improves the tracking performance by simultaneously attending to a larger visual field and retaining more raw information with high resolution. Inspired by the HVS data processing with limited resources, our method formulates the non-uniform resizing as an explicitly controlled magnification of important areas with restriction on extreme deformation. Extensive experiments have demonstrated that our method can bridge the gap between speed-oriented and performance-oriented trackers with negligible computational overhead.

**Limitations.** One limitation of our method is that we need a fixed context factor to determine the size of the visual field. It would be interesting to develop a method to dynamically adjust the size of the visual field according to the tracking trajectory or appearance of the target in future work.

**Acknowledgements.** The authors would like to thank the anonymous reviewers for their valuable comments and suggestions. This work was supported in part by the National Key R&D Program of China (Grant No. 2020AAA0106800), the Natural Science Foundation of China (Grant No. U22B2056, 61972394, 62036011, 62192782, 61721004, U2033210), the Beijing Natural Science Foundation (Grant No. L223003, JQ22014, 4234087), the Major Projects of Guangdong Education Department for Foundation Research and Applied Research (Grant No. 2017KZDXM081, 2018KZDXM066), the Guangdong Provincial University Innovation Team Project (Grant No. 2020KCXTD045). Jin Gao and Bing Li were also supported in part by the Youth Innovation Promotion Association, CAS.

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

# Appendices

## A    Derivations of the QP-based Minimization Task for Grid Manipulation

The minimization task for grid manipulation in our non-uniform resizing is defined as

$$\underset{d_l^{row}, \, d_k^{col}}{\text{minimize}} \quad E = E_{zoom} + \lambda E_{rigid}$$
$$\text{subject to} \quad \sum_{l=1}^{m} d_l^{row} = H, \ \sum_{k=1}^{n} d_k^{col} = W \tag{A1}$$

where the zoom energy $E_{zoom}$ is defined as

$$E_{zoom} = \sum_{l=1}^{m} \sum_{k=1}^{n} S_{k,l}^2 \left( \left( d_l^{row} - \frac{1}{\gamma} \frac{H}{m} \right)^2 + \left( d_k^{col} - \frac{1}{\gamma} \frac{W}{n} \right)^2 \right) , \tag{A2}$$

and the rigid energy $E_{rigid}$ is defined as

$$E_{rigid} = \sum_{l=1}^{m} \sum_{k=1}^{n} S_{k,l}^2 \left( \frac{m}{H} d_l^{row} - \frac{n}{W} d_k^{col} \right)^2 . \tag{A3}$$

This minimization task in Eq. (A1) can be efficiently solved using a standard QP solver [20], which requires converting the above minimization task into the general form of QP, *i.e.*,

$$\underset{d}{\text{minimize}} \quad E = \frac{1}{2} d^\top P d + q^\top d$$
$$\text{subject to} \quad A d = b \tag{A4}$$

where $d = (d_1^{row}, ..., d_m^{row}, d_1^{col}, ..., d_n^{col})^\top \in \mathbb{R}^{m+n}$ is the unknown grid intervals to be optimized. $P \in \mathbb{R}^{(m+n) \times (m+n)}$ and $q \in \mathbb{R}^{m+n}$ can be derived from the energy function $E = E_{zoom} + \lambda E_{rigid}$. $A \in \mathbb{R}^{2 \times (m+n)}$ and $b \in \mathbb{R}^2$ can be derived from the linear constraint in Eq. (A1).

**Converting the energy function into the matrix form.** Denote $\mathcal{S}_k^{col} = \sum_{l=1}^{n} S_{k,l}^2$, and $\mathcal{S}_l^{row} = \sum_{k=1}^{m} S_{k,l}^2$, the overall energy $E$ can be expanded as

$$E = \sum_{l=1}^{m} \mathcal{S}_l^{row} \left( \lambda \left( \frac{m}{H} \right)^2 + 1 \right) d_l^{row2} + \sum_{k=1}^{n} \mathcal{S}_k^{col} \left( \lambda \left( \frac{n}{W} \right)^2 + 1 \right) d_k^{col2}$$
$$- \sum_{l=1}^{m} 2 \mathcal{S}_l^{row} \frac{H}{\gamma m} d_l^{row} - \sum_{k=1}^{n} 2 \mathcal{S}_k^{col} \frac{W}{\gamma n} d_k^{col} \tag{A5}$$
$$- \sum_{l=1}^{m} \sum_{k=1}^{n} 2 \lambda S_{k,l}^2 \frac{mn}{HW} d_l^{row} d_k^{col} + C$$

where $C$ is a constant term that can be ignored during minimization. The first, second and third lines of Eq. (A5) are consisting of squared terms, linear terms and cross terms respectively.

Thus, the overall energy $E$ can be converted into the matrix form described in Eq. (A4) by fitting the above terms into $P \in \mathbb{R}^{(m+n) \times (m+n)}$ and $q \in \mathbb{R}^{m+n}$, *i.e.*,

$$P = \begin{pmatrix} P^{row} & P^{cross} \\ (P^{cross})^\top & P^{col} \end{pmatrix}, \quad q = \begin{pmatrix} q_{row} \\ q_{col} \end{pmatrix} \tag{A6}$$

where $P^{row} \in \mathbb{R}^{n \times n}$ and $P^{col} \in \mathbb{R}^{m \times m}$ are diagonal matrices accounting for the squared terms in $E$. $P^{row}$ and $P^{col}$ can be defined as

$$P_{k,l}^{row} = \begin{cases} 2 \mathcal{S}_l^{row} (\lambda (\frac{m}{H})^2 + 1), k = l \\ 0, others \end{cases} , \quad P_{k,l}^{col} = \begin{cases} 2 \mathcal{S}_k^{col} (\lambda (\frac{n}{W})^2 + 1), k = l \\ 0, others \end{cases} . \tag{A7}$$

$P^{cross} \in \mathbb{R}^{m \times n}$ is a matrix accounting for the cross terms in $E$, while $q^{row} \in \mathbb{R}^n$ and $q^{col} \in \mathbb{R}^m$ account for the linear terms. $P^{cross}$, $q^{row}$ and $q^{col}$ can be defined as

$$P_{k,l}^{cross} = -2\lambda S_{k,l}^2 \frac{mn}{HW}, \quad q_l^{row} = -2\mathcal{S}_l^{row} \frac{H}{\gamma m}, \quad q_k^{col} = -2\mathcal{S}_k^{col} \frac{W}{\gamma n} . \tag{A8}$$

**Converting the linear constraint into the matrix form.** The linear constraint can fit in the matrix form described in Eq. (A1) by defining $A \in \mathbb{R}^{2 \times (m+n)}$ and $b \in \mathbb{R}^2$ as

$$A_{k,l} = \begin{cases} 1, k = 1, l = 1, ..., m \\ 1, k = 2, l = m + 1, ..., n + n \\ 0, others \end{cases} , \quad b = (H, W)^\top . \tag{A9}$$

**Proof of the convexity.** When the matrix $P$ in Eq. (A6) is positive semi-definite, the energy function in Eq. (A1) is convex and a global minimization can be found. Apparently, $P$ is positive semi-definite since, for any $d \in \mathbb{R}^{m+n}$,

$$
\begin{aligned}
d^\top P d =& 2\left( \sum_{l=1}^m \mathcal{S}_l^{row} \left( \lambda \left( \frac{m}{H} \right)^2 + 1 \right) d_l^{row\,2} + \sum_{k=1}^n \mathcal{S}_k^{col} \left( \lambda \left( \frac{n}{W} \right)^2 + 1 \right) d_k^{col\,2} \right. \\
& \left. - \sum_{l=1}^m \sum_{k=1}^n 2\lambda S_{k,l}^2 \frac{mn}{HW} d_l^{row} d_k^{col} \right) \\
=& 2\left( \sum_{l=1}^m \mathcal{S}_l^{row} d_l^{row\,2} + \sum_{k=1}^n \mathcal{S}_k^{col} d_k^{col\,2} + \lambda \sum_{l=1}^m \sum_{k=1}^n S_{k,l}^2 \left( \frac{m}{H} d_l^{row} - \frac{n}{W} d_k^{col} \right)^2 \right) \\
\geq& 0
\end{aligned}
\tag{A10}
$$

# B    Attribute-based Analysis on LaSOT

We further analyze the AUC gains brought by our method on the two baselines, *i.e.*, OSTrack and TransT, based on the attributes of the videos on LaSOT [11]. The results are displayed in Fig. A1.

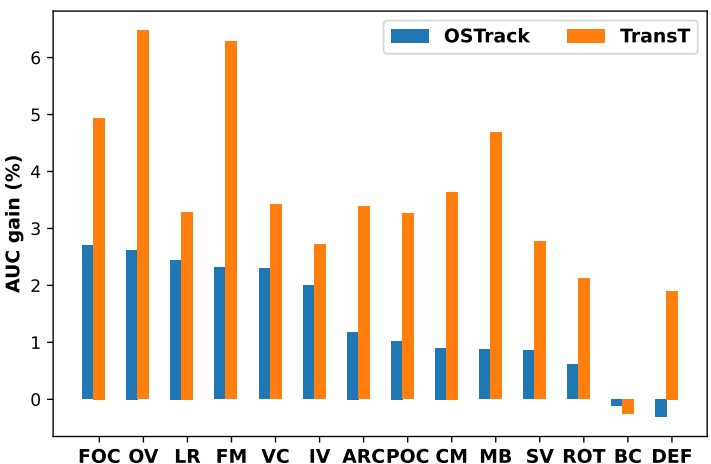

Figure A1: Attribute-based analysis of AUC gains on LaSOT [11] [3]

As shown in Fig. A1, our method significantly improves the performance of both baselines under challenging scenarios with drastic movement (see Fast motion (FM), Viewpoint change (VC), *etc*) or

---

[3]FOC: Full Occlusion, OV: Out-of-view, LR: Low Resolution, FM: Fast Motion, VC: Viewpoint Change, IV: Illumination Variation, ARC: Aspect Ration Change, POC: Partial Occlusion, CM: Camera Motion, MB: Motion Blur, SV: Scale Variation, ROT: Rotation, BC: Background Clutter, DEF: Deformation

significant drift after some period of time (see Full Occlusion (FOC), Out-of-View (OV), *etc*), owning to the enlarged visual field. Performance on videos with low resolution (LR) is also considerably improved thanks to the zoom-in behavior. The deformation caused by non-uniform resizing may cause a little performance degradation when encountering background clutter (BC) or deformation (DEF).

## C  Comparison Between Non-uniform Resizing and Optimized Uniform Resizing Baseline

To demonstrate the superiority of our method, we additionally explore whether optimizing the crop hyper-parameters in the uniform resizing baseline can have a similar effect to our proposed non-uniform resizing, the results are shown in Tab. A1.

Despite the fact that our limited compute resources make us unable to optimize the above hyper-parameters on large tracking datasets in a grid-search manner, our adequately powered experiments clearly show that the above optimizing is indeed hardly able to achieve similar effects to our resizing. Even for the baseline OSTrack-256[33] with the optimal crop hyper-parameters on LaSOT[11] (69.5 AUC), we experimentally find that it can only achieve AUC gains of +0.4 and +0.6 on LaSOT$_{ext}$[12] and TNL2K[30] respectively while performing worse than the original OSTrack-256 on TrackingNet[23] (83.1 SUC vs 82.1 SUC). In other words, our method still outperforms the above optimal baseline on LaSOT (+0.7 AUC), LaSOT$_{ext}$ (+2.7 AUC), TNL2K (+1.6 AUC), TrackingNet (+1.1 SUC). Moreover, it is experimentally shown that the mismatch of the object size between the search and template images caused by increasing the search-image context factor has a relatively small effect on the final tracking accuracy for modern transformer trackers. That means the low resolution of the object is the major cause for the worse accuracy.

Table A1: Comparison results of our method and the uniform resizing baseline with its crop hyper-parameters varied (*i.e.*, setting $c^w$ and $c^h$ using two different choices as in Sec. 3.1, varying the search-image context factor, and varying the template-image size to alleviate the mismatch of the object size between the search and temple images). The experiments are based on OSTrack-256[33] and tested on LaSOT[11]. The context factor and size of search/template images are displayed as `context factor/size`. The choices of $c^w, c^h$: ❶ $c^w = b^w, c^h = b^h$ and ❷ $c^w = c^h = (b^w + b^h)/2$

|  | Original | $c^w, c^h$ | Object Size Mismatched | | | | Object Size Matched | | | | Ours |
|---|---|---|---|---|---|---|---|---|---|---|---|
| Search | 4/256 | 4/256 | 4.5/256 | 5/256 | 5.5/256 | 6/256 | 4.5/256 | 5/256 | 5.5/256 | 6/256 | 5/256 |
| Template | 2/128 | 2/128 | 2/128 | 2/128 | 2/128 | 2/128 | 2/114 | 2/102 | 2/93 | 2/85 | 2/128 |
| $c^w, c^h$ | ❶ | ❷ | ❶ | ❶ | ❶ | ❶ | ❶ | ❶ | ❶ | ❶ | ❶ |
| AUC | 69.1 | 68.5 | 69.0 | 69.5 | 68.2 | 67.5 | 69.1 | 68.8 | 68.0 | 67.4 | **70.2** |

## D  More Experimental Results on UAV123

In this section, we report the performance comparison on an additional benchmark, *i.e.*, UAV123 [22]. UAV123 is an aerial tracking benchmark captured from low-altitude UAVs with more small objects and larger variations in bounding box size and aspect ratio. As shown in Tab. A2, our method achieves 1.0% and 0.6% relative gains on AUC for the OSTrack [33] and TransT [8] baselines, demonstrating that our method can improve the tracking performance in a wide range of tracking scenarios.

Table A2: Comparison with state-of-the-art trackers on UAV123[22]

| Dataset | OSTrack-Zoom | OSTrack [33] | TransT-Zoom | TransT [8] | MixFormer-L [9] | ToMP [21] | DiMP [4] | SiamFC [3] |
|---|---|---|---|---|---|---|---|---|
| AUC (%) | 69.3 | 68.3 | 69.7 | 69.1 | 69.5 | 69.0 | 65.3 | 37.7 |

# E   More Experimental Results with SwinTrack as the Baseline

We further apply our method to both the performance-oriented version SwinTrack-B (backbone: Swin Transformer-Base) and speed-oriented version SwinTrack-T (backbone: Swin Transformer-Tiny) of SwinTrack [19] to experimentally show the generalization capability of our proposed resizing method for a wide range of visual tracking algorithms. Note that we use the v1 version of SwinTrack (without motion token) as the baseline because only the code of v1 version is publicly available.

As shown in Tab. A3, our method can bring consistent improvement for both SwinTrack-T and SwinTrack-B. Particularly worth mentioning is that our SwinTrack-B-Zoom can even outperforms SwinTrack-B-384 with a much smaller input size.

Table A3: Further validation of applying our non-uniform resizing to both the performance-oriented version (backbone: Swin Transformer-Base) and speed-oriented version (backbone: Swin Transformer-Tiny) of SwinTrack [19]. Note that we use the v1 version of SwinTrack (without motion token) as the baseline because only the code of v1 version is publicly available.

| Type | Trackers | Size | LaSOT[11] | |
|---|---|---|---|---|
| | | | AUC | P |
| Speed-oriented | SwinTrack-T-Zoom | 224 | **68.5** | **72.9** |
| | SwinTrack-T[19] | 224 | 66.7 | 70.6 |
| Performance-oriented | SwinTrack-B-Zoom | 224 | **70.5** | **75.4** |
| | SwinTrack-B[19] | 224 | 69.6 | 74.1 |
| | SwinTrack-B-384[19] | 384 | 70.2 | 75.3 |

# F   Analysis on Small Improvement Over TrackingNet

To find out the reason for the small improvement over TrackingNet [23], we additionally analyze the ratios of challenging scenarios appearing in the different tracking datasets (see Tab. A4), which clearly show that the test split of TrackingNet has a very different property from LaSOT [11], $LaSOT_{ext}$ [12], and TNL2K [30] with respect to the ratios of challenging scenarios. As shown in Fig. A1, our method can achieve significant performance gains under challenging scenarios with drastic movement (see Fast Motion (FM), Viewpoint Change (VC), *etc.*) or significant drift after some period of time (see Full Occlusion (FOC), Out-of-View (OV), *etc.*) owning to the enlarged visual field. However, the ratios of the challenging scenarios FOC, OV, FM in TrackingNet (VC is not labeled in TrackingNet) are significantly lower than in LaSOT, $LaSOT_{ext}$, and TNL2K, which may be the reason for small improvement of our approach on TrackingNet.

Table A4: Ratios of challenging scenarios appearing in the different tracking datasets. Our method can achieve significant performance gains under challenging scenarios with drastic movement (*e.g.*, Fast Motion (FM), Viewpoint Change (VC)) or significant drift after some period of time (*e.g.*, Full Occlusion (FOC), Out-of-View (OV)) owning to the enlarged visual field. Note that VC is not labeled in TrackingNet.

| Datasets | Ratios of Challenging Scenarios | | | | AUC Gains | |
|---|---|---|---|---|---|---|
| | Significant Drift | | Drastic Motion | | OSTrack[33] | TransT[8] |
| | FOC | OV | FM | VC | | |
| LaSOT[11] | 42.1% | 37.1% | 18.9% | 11.8% | +1.1% | +2.2% |
| $LaSOT_{ext}$[12] | 62.7% | 21.3% | 58.7% | 39.3% | +3.1% | +2.0% |
| TNL2K[30] | 13.9% | 33.4% | 24.0% | 44.3% | +2.2% | +3.0% |
| TrackingNet[23] | 4.7% | 4.0% | 10.0% | - | +0.1% | +0.4% |

## G  Detailed Latency of Non-uniform Resizing

We provide the latency analysis in terms of time (ms) and MACs for solving QP and resizing images in Tab. A5. It can be seen that resizing images with interpolations in our method costs about half of the total time in spite of its parallel processing, while iteratively solving QP costs the rest of the total time despite its much lower FLOPs.

Table A5: Detailed latency of non-uniform resizing

| Component | Time(ms) | MACs |
|---|---|---|
| Solving QP | 0.89 | 0.16M |
| Resizing | 0.69 | 1.12M |
| Total | 1.58 | 1.28 M |

## H  Confidence Interval for Ablation Experiments of Hyper-parameters

We calculate the 90% confidence intervals for all the experiments in Tab. 3 using bootstrap sampling. Each of the 90% confidence intervals is obtained using 10000 bootstrap samples, each sample with size 280 (same as the video number of LaSOT-test-split). The results are listed in Tab. A6.

Table A6: 90% confidence intervals for ablation experiments of hyper-parameters.

(a) Bandwidth $\beta$

| $\beta$ | 4 | 64 | 256 |
|---|---|---|---|
| Upper | 70.95 (+1.87) | 72.02 (+1.86) | 71.54 (+1.90) |
| Mean | 69.08 | 70.16 | 69.64 |
| Lower | 67.23 (-1.85) | 68.33 (-1.83) | 67.80 (-1.84) |

(b) Control grid size

| Size | 8 | 16 | 32 |
|---|---|---|---|
| Upper | 68.50 (+2.06) | 72.02 (+1.86) | 71.52 (+1.98) |
| Mean | 66.44 | 70.16 | 69.54 |
| Lower | 64.49 (-1.95) | 68.33 (-1.83) | 67.63 (-1.91) |

(c) Zoom factor $\gamma$

| $\gamma$ | 1.25 | 1.5 | 1.75 |
|---|---|---|---|
| Upper | 71.58 (+1.89) | 72.02 (+1.86) | 71.35 (+1.94) |
| Mean | 69.69 | 70.16 | 69.41 |
| Lower | 67.88 (-1.81) | 68.33 (-1.83) | 67.49 (-1.92) |

(d) Balance factor $\lambda$

| $\lambda$ | 0 | 1 | 2 |
|---|---|---|---|
| Upper | 71.69 (+1.89) | 72.02 (+1.86) | 71.43 (+1.87) |
| Mean | 69.80 | 70.16 | 69.56 |
| Lower | 67.93 (-1.87) | 68.33 (-1.83) | 67.71 (-1.85) |

## I  More Visualization Results

We visualize the results of our method OSTrack-Zoom and its corresponding baseline on videos with challenging attributes to qualitatively show the superiority of our method in Fig. A2. Thanks to the zoom-in behavior and the enlarged visual field, our method can better handle the challenges such as full occlusion, fast motion, *etc*.

As shown in the left part of Fig. A2, at frame #150, both our method and the baseline lose track of the target since the target is fully occluded. At frame #157, our method quickly re-detects the target as the target re-appears. Note the target is significantly larger in the resized input image when using our resizing method, which indicates that the zoom-in behavior allowed by our method helps the re-detection of the target. At frame #187, our method successfully tracks the target whereas the baseline still drifts to a similar car.

As shown in the right part of Fig. A2, at frame #15, the target is moving really fast. As a result, in the resized input image generated by the baseline, part of the target is out of view which causes an inaccurate localization. In contrast, the resized input image generated by our method keeps the entire target visible thanks to the enlarged visual field, resulting in a more accurate tracking result. The situation is similar in frame #27, where the baseline tracks a fully-preserved distractor rather than

the half-cropped target, whereas our method successfully tracks the target since the target is intact. At frame #32, the baseline completely lose track of the target, while our method successfully tracks it.

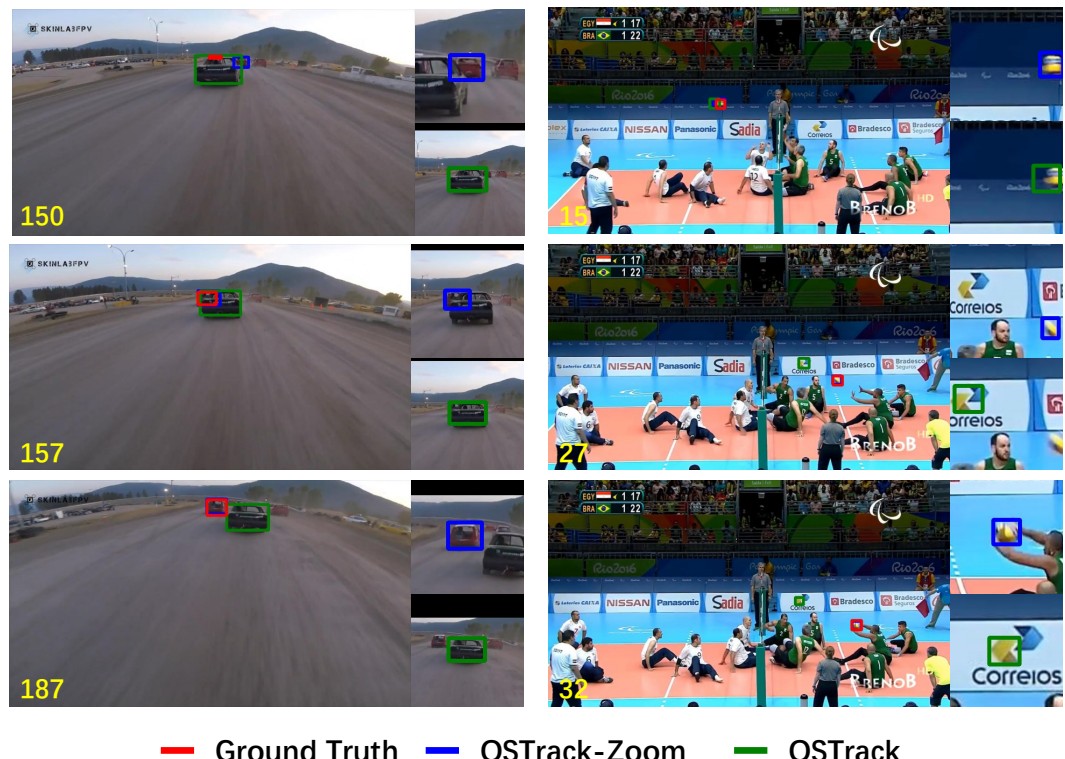


━━ **Ground Truth**    ━━ **OSTrack-Zoom**    ━━ **OSTrack**


Figure A2: Visualization of tracking results on videos with challenging attributes. Input to the tracker is displayed on the right side of each full image, where the image with blue box is the input image resized by our method and the image with green box is the input image resized by the baseline. (*left*) An example of video with full occlusion. The zoom-in behavior allows our method to quickly re-detect the target and avoid drifting. (*right*) An example of video with fast motion. The enlarged visual field allows our tracker to have the whole target visible without cropping, which facilitates localization and avoids drifting to similar objects when the target has drastic motion.

## J    Failure Cases

We visualize the failure cases of our method by comparing the tracking results of OSTrack-Zoom (blue) and OSTrack (green) in Fig. A3.

As shown in the left part of Fig. A3, at frame #1108, the target is out of view. As our method enlarges the visual field, apart from the distractor (orange ball) at the center of the input, an additional distractor (dark green ball at the top-right of the input image) comes into view, which causes the tracker to predict a large box between the two distractors. The localization error quickly accumulates to an overly enlarged visual field at frame #1117, since the visual field is partly determined by the previous prediction. At this time, a white ball that is very similar to the template is brought into view, causing our method to drift to the far-away white ball. Then, at frame #1130, when the target re-appears, our method cannot re-detect the target as our method drifts to a far-away distractor, which makes the real target out of the visual field of our tracker.

As shown in the right part of Fig. A3, at frame #94, both our method and the baseline successfully track the target. However, at frame #104, the enlarged visual field from our method brought additional contexts (orange robotic arm on the left) into view. The little fraction of orange robotic arm on the left makes the distractor look like a horizontally flipped template which has a small fraction of orange robotic arm on the right. As a result, our method drifts to the distractor. Then, at frame #114, our

method fails to track back to the squeezed real target since it is at the edge of the input image and partly invisible.

From the visualization of the failure cases, we find that, although an enlarged visual field promotes tracking performance in most cases, sometimes additional distractors or contexts brought by the fixed large visual field will cause drifting. Thus, it would be interesting to develop a method to dynamically adjust the visual field based on the surrounding of the target and its tracking history in the future.

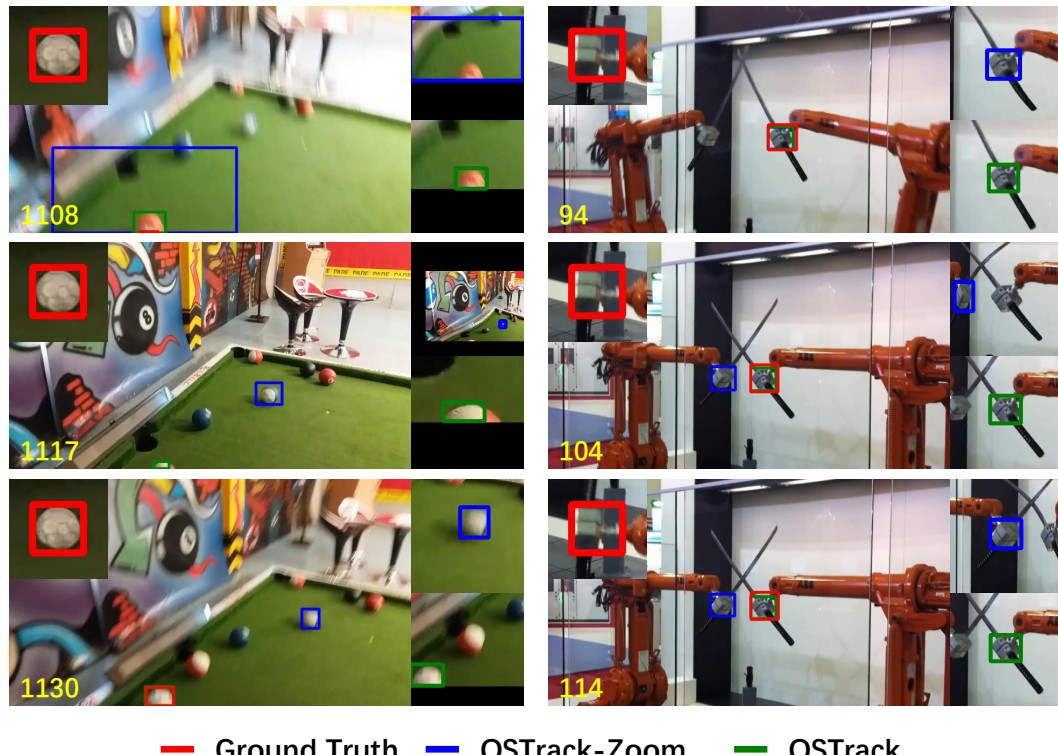

**━ Ground Truth**   **━ OSTrack-Zoom**   **━ OSTrack**

Figure A3: Visualization of failure cases. Input to the tracker is displayed on the right side of each full image, where the image with blue box is the input image resized by our method and the image with green box is the input image resized by the baseline. Template image is visualized on the top-left corner of the full image. (*left*) A combination of out-of-view and fast motion causes an accumulated prediction error which leads to drifting. Localization with an enlarged visual field under some challenging scenarios may be difficult and its wrong prediction may produce less desirable results since the enlarged visual field brought more distractors. The localization error quickly accumulates to an overly enlarged visual field which then causes drifting. (*right*) Cluttered background with similar distractors causes drifting. The additional context (orange robotic arm) brought by the enlarged visual field makes the distractor on the left look like a horizontally flipped template, which causes drifting.

