# OpenReview forum: "ZoomTrack: Target-aware Non-uniform Resizing for Efficient Visual Tracking"
_NeurIPS.cc/2023/Conference — NeurIPS 2023 spotlight_

### Official Review · Reviewer_2uk1 · 2023-07-03

**Soundness:** 3 good
**Presentation:** 3 good
**Contribution:** 4 excellent
**Rating:** 7
**Confidence:** 5

**Summary:**

The paper studies a very interesting problem in local visual tracking with crop-then-resize operations: if it is possible to reduce the high computational demand of attending to a large visual field while retaining its potential benefit of achieving SOTA tracking accuracy. To this end, the paper takes inspiration from the processing of the human retina and proposes an elegant non-uniform resizing method to enable retaining more raw information for the target area while resizing a cropped image with a large visual field to a smaller size. By enforcing an explicitly controlled magnification of important areas with restriction on extreme deformation on top of a grid representation, this non-uniform resizing can be achieved efficiently based on QP. The authors select two representative local trackers (TransT and OSTrack) to demonstrate the well-balanced tracking accuracy and speed of trackers that are based on the proposed resizing module. Extensive experiments are conducted on five challenging tracking benchmarks, which clearly show the superiority of the proposed approach.

**Strengths:**

In general, I am positive about this paper. I am glad to see this paper paying some effort to improve the little-explored cropping and resizing strategy in local visual trackers. The strengths of this paper are as follows:

1, Good organization and writing. The paper is well organized and the flow is smooth.

2, Novelty. The proposed non-uniform resizing method is quite interesting and looks solid to me, which enables retaining more raw information for the target area during resizing.

3, Promising performance. This paper enables OSTrack to achieve 70.2% AUC on LaSOT with 100fps, which is impressive. Most existing competitors always rely on heavy trackers to achieve AUC score above 70% on LaSOT. The performance gap between the proposed efficient tracker and the heavy trackers on GOT-10k, LaSOT_ext, and TNL2K is even closed. The experimental analyses are also aligned with the motivation of attending to a large visual field.

**Weaknesses:**

1, Only two transformer trackers are selected for experimental validations. One is the hybrid CNN-Transformer tracker TransT and the other one is the one-stream Transformer tracker OSTrack. It would be better if one more purely CNN-based tracker or some other top performer is selected to further show the generalization capability. The conclusion would be more convincing.

2, In Table 1, it is observed that the improvement on TrackingNet is very small. The SUC performance gain is only 0.1%. I suppose there are some detailed explanations for the reasons but obviously, there is not.

3, It is not clear to me whether the proposed non-uniform resizing is also applied to the cropped template. A detailed analysis in this regard is also required to make the paper self-contained and fully reproducible.

**Questions:**

There are several questions.

1, In line 198-199, the authors said they use a (m+1)*(n+1) small-sized controllable grid to approximate the grid G with shape h*w. How can a grid be approximated by another grid with a different size?

2, In Eq. (6) for determining the important score of different areas, a small constant is also used to prevent extreme deformation. What’s the relationship between this part and the specially designed rigid energy in Eq. (8) following G2?

3, In Table 1 and Figure 1, the authors use MACs to indicate the computational overhead, while in line 353 they use FLOPS. Please explain this inconsistency. Moreover, is the computational overhead of your method in Table 1 exactly the same as the corresponding baseline?

Overall, I like this paper a lot, including the novelty with broad interest and its promising performance. However, I rated it as weak accept (6) now due to the above weaknesses and the above-mentioned questions. If my concerns and questions are addressed sufficiently, I will increase my score.

**Limitations:**

Yes, the limitations have been briefly addressed in the paper.

---

> ### Author Rebuttal · Authors · 2023-08-09
>
> Thank you for your review and feedback! We are happy that you find our method quite interesting and solid, which studies a little-explored problem, and acknowledge that the performance gain on LaSOT is impressive and the experiments clearly show the superiority of our elegant solution. We provide responses to the specific points below:
>
> **About more experimental validations to further show the generalization capability**: We further apply our method to both the performance-oriented version (backbone: Swin Transformer-Base) and speed-oriented version (backbone: Swin Transformer-Tiny) of SwinTrack [15] to experimentally show the generalization capability of our proposed resizing method for wide range of visual tracking algorithms. Due to the limited computing resources, we only report the comparisons on the large-scale LaSOT dataset (see below), which further clearly shows the superiority of our approach. Note that we use the v1 version of SwinTrack as the baseline because only the code of v1 version is publicly available. We will include this new baseline in the revision.
> ||Size| AUC|P|
> |-|-|-|-|
> |**Speed-oriented**||||
> |SwinTrack-T-Zoom|224|**68.5**|**72.9**|
> |SwinTrack-T|224|66.7|70.6|
> |**Performance-oriented**||||
> |SwinTrack-B-Zoom|224|**70.5**|**75.4**|
> |SwinTrack-B| 224|69.6|74.1|
> |SwinTrack-B-384|384|70.2|75.3|
>
> **About the reason for small improvements on TrackingNet**: We additionally analyze the ratios of challenging scenarios appearing in the different tracking datasets (see below), which clearly show that the test split of TrackingNet has a very different property from LaSOT, LaSOT_ext, and TNL2K with respect to the ratios of challenging scenarios. As shown in Fig. S1 of our appendix material, our method can achieve significant performance gains under challenging scenarios with drastic movement (see Fast motion (FM), Viewpoint change (VC)) or significant drift after some period of time (see Full Occlusion (FOC), Out-of-View (OV)) owning to the enlarged visual field. However, the ratios of the challenging scenarios FOC, OV, FM in TrackingNet (VC is not labeled in TrackingNet) are significantly lower than in LaSOT, LaSOT_ext, and TNL2K, which may be the reason for small improvements of our approach on TrackingNet. We will include this new analysis in the revision.
> |Dataset|Ratio of FOC|Ratio of OV|Ratio of FM|Ratio of VC|AUC Gain (OSTrack)|
> |-|-|-|-|-|-|
> |LaSOT|42.1%|37.1%|18.9%|11.8%|+1.1%|
> |LaSOT_ext| 62.7%| 21.3%| 58.7%| 39.3%| +3.1%|
> |TNL2K| 13.9%|33.4%| 24.0%| 44.3%| +2.2%|
> |TrackingNet|4.7%|4.0%|10.0%|- (not labeled)|+0.1%|
>
> **About the confusion of applying the non-uniform resizing to the cropped template**: We are sorry for not clearly indicating that we did not apply our non-uniform resizing on the template image. Although the template image can be non-uniformly resized by solving QP with the ground truth annotation and a relatively small context factor only once at the beginning frame during testing, we experimentally find that either applying non-uniform resizing on template image alone or on both template and search images performs worse than applying it only on search image (see below). We think the reason is that the template image must have a relatively small context factor to contain minimal necessary context, which makes the whole template region important for the tracking network to locate the target. So the trackers cannot benefit from non-uniform resizing of template images. We will include this new analysis in the revision.
> ||Zoom|Zoom|LaSOT|LaSOT|
> |-|-|-|-|-|
> ||Template|Search|AUC|P|
> |OSTrack-256|√||64.8|71.0|
> |OSTrack-256 ||√|**70.2**|**76.2**|
> |OSTrack-256 |√|√| 69.6|75.5|
>
> **About the confusion of one grid approximated by another grid**: We are sorry for this misleading expression. We can use "estimated" instead of "approximated" here, i.e., the larger grid is estimated by interpolating the smaller grid. We will correct it in the revision.
>
> **About the relationship between the small constant in Eq. (6) and the rigid energy in Eq. (8)**: Thanks for the thoughtful comment. On the one hand, the small constant ensures importance score $S_{k,l}\neq 0$. Thus, every location $(k,l)$ cannot have extreme $d_l^{row}$ and $d_k^{col}$, otherwise the total energy will be high. On the other hand, the rigid energy further prevents extreme aspect ratio ($d_k^{col} / d_l^{row}$) change, which is also imposed on every location because the importance score $S_{k,l}\neq 0$. In other words, both the small constant and rigid energy help to avoid extreme deformation. We will give a clearer explanation in the revision.
>
> **About the inconsistency of using MACs and FLOPs and the confusion of computational overhead in Tab. 1**: We are sorry for the misleading expression. **First**, it is a common practice to use MACs to evaluate the computational overhead of a deep neural network. We follow this common practice and use MACs to compare with other methods. However, our non-uniform resizing contains non-pytorch operations, e.g., solving the QP problem, therefore it cannot be measured by the commonly used counter 'pytorch-OpCounter' designed for counting pytorch operations. We thus use the Performance Application Programming Interface (PAPI) to calculate the CPU FLOPs for our method. Generally speaking, 1 MACs approximately equals 2 FLOPs (1 multiply and 1 accumulation operations). The FLOPS for our method to resize an image is 2.56M FLOPs which is approximately equal to 1.28M MACs. Thanks for pointing out this inconsistency. We will convert the FLOPs to MACs in the final version for consistency. **Second**, we indeed add the additional overhead (1.28M MACs) to the corresponding baseline in Tab. 1 so they are not the same. As the additional overhead is far less than the overhead of the network inference (1.28M MACs v.s. 21.5G MACs), the difference is rounded off. Thanks for pointing it out. We will make an explanation in the revision.

---

> > ### Comment · Reviewer_2uk1 · 2023-08-16
> > **Thank authors for the reply**
> >
> > I greatly appreciate the authors’ efforts in the rebuttal. I think my concerns have been successfully addressed, such as the additional validation for generalizing to other trackers, the clarification on the small improvement on TrackingNet, the used hyper-parameters, and the implementation of not applying ZoomTrack to the template. I am also willing to increase my score to Accept after reading the feedback and the other reviews (which I think are unanimously as favorable as mine regarding the motivation and technical solution). I hope the authors will incorporate the additional results and analyses into the revised paper.

---

> > > ### Author Response · Authors · 2023-08-17
> > > **Thank Reviewer 2uk1 for the reply**
> > >
> > > We are grateful for your increased score to Accept. We will incorporate the additional results and analyses following your suggestions and acknowledge the anonymous reviewers in the revised version of our paper.

---

### Official Review · Reviewer_P4Ao · 2023-07-04

**Soundness:** 3 good
**Presentation:** 2 fair
**Contribution:** 2 fair
**Rating:** 6
**Confidence:** 5

**Summary:**

The authors propose a non-uniform resizing method for visual tracking, where they aim to boost the performance of transformer-based visual tracking methods at negligible additional computational cost. They achieve this by reducing the input size while attending to the relevant parts of the target, and this approach is applicable to wide range of crop-based tracking algorithms. They show the results of their algorithm on various large-scale datasets.

**Strengths:**

- Authors focus on the easily overlooked aspect of visual tracking algorithms, and their approach seems technically plausible while bringing adequate enlightenment to the field.

- Their approach shows competitive performance on large-scale benchmarks compared to contemporary tracking algorithms.

- They validate their method on multiple baseline algorithms with ablation experiments.

**Weaknesses:**

- Although the motivation of the proposed method seems plausible, there seems to be a lack of validation when it comes to analyzing the importance of choosing input resizing method for trackers. Are there further analysis on the impact of input resizing for wide range of visual tracking algorithms?

- Since the formulation for the rectangular grid is hand-crafted using multiple controlallable hyperparameters (e.g. bandwitdh of Gaussians $\beta$, choice of $\epsilon, \gamma, \lambda$, ...), results in Table 3 shows some sensitivity to these hyperparameter choices. Considering these results and the results in Table 1, the performance gains obtained by the proposed strategy seems limited.

- What are the reasons for the design choices of choosing (1) Gaussian importance scores for target area, (2) Using quadratic formulation for zoom and rigid energy? Are there alternative formulations tested?

- Proposed non-uniform grid formulation seems to be target content (e.g. target object class, color, etc.) agnostic and only dependent on the input bounding box position and size. Since a simple learning based model (e.g. lightweight MLP) is also available, would this have been a better approach?

**Questions:**

Please refer to the weaknesses section for details. I generally concur with the motivations raised by the authors, but their approach to the problem needs more explanation and justifications for their design choices are needed.

**Limitations:**

Authors provide a limitations section at the end of their paper, and the limitations are adequately addressed.

---

> ### Author Rebuttal · Authors · 2023-08-09
>
> Thank you for your review and feedback! We are happy that you find our method technically plausible while bringing adequate enlightenment to the visual tracking field by studying an easily overlooked problem. We provide responses to the specific points below:
>
> **About more experimental validations and analyses to further show the importance of choosing input resizing method**: We further apply our method to both the performance-oriented version (backbone: Swin Transformer-Base) and speed-oriented version (backbone: Swin Transformer-Tiny) of SwinTrack [15] to experimentally show the generalization capability of our proposed resizing method for wide range of visual tracking algorithms. Due to the limited computing resources, we only report the comparisons on the large-scale LaSOT dataset (see below), which further clearly show the superiority of our approach. Note that we use the v1 version of SwinTrack as the baseline because only the code of v1 version is publicly available. Although analyzing the importance of choosing input resizing method is an overlooked and little-explored problem in the visual tracking field and the common practice is using uniform resizing by default, we believe our further experimental analysis based on SwinTrack, as well as the previously explored two baselines (OSTrack and TransT), can demonstrate the importance of choosing appropriate input resizing method for trackers. We will include this new baseline in the revision.
>
> || Size | AUC  | P    |
> |-|-|-|-|
> |**Speed-oriented**|      |      |      |
> | SwinTrack-T-Zoom| 224  |**68.5**|**72.9**|
> | SwinTrack-T| 224  | 66.7 | 70.6 |
> | **Performance-oriented** |      |      |      |
> | SwinTrack-B-Zoom| 224  |**70.5**|**75.4**|
> | SwinTrack-B| 224  | 69.6 | 74.1 |
> | SwinTrack-B-384| 384  | 70.2 | 75.3 |
>
> **About how the hyper-parameters in our proposed method are tuned and how they work**: We claim that the small set of hyper-parameters (acknowledged by Reviewer `Gq4E`) in our QP formulation based on grid representation (e.g., $\beta$, $\gamma$, $\lambda$, grid size) are tightly coupled with our formulation and also have very clear meanings in our framework, so that we can choose a set of reasonable initial values (which may be the optimal values) and then conduct small (coordinate-wise) sweep (acknowledged by Reviewer `Gq4E`) for them to test the sensitivity of our OSTrack-Zoom to these hyper-parameter choices in Tab. 3. For example, by design, we make the zoom factor $\gamma$ capable of controlling the amount of magnification. The performance-oriented version of OSTrack uses 1.5$\times$ larger search-image size than its speed-oriented version, so it is natural to set the initial value of $\gamma$ to 1.5 in our OSTrack-Zoom, which proved to be an appropriate value in our experiments. Another example is the control grid size. It is natural to use 16$\times$16 grid because the search image after non-uniform resizing is evenly split into 16$\times$16 patches and then sent into the Vision Transformer (ViT) backbone for feature extraction. As our resizing process can be intuitively understood as resizing each rectangular grid patch on the source image to the one on the target image, we can expect that each patch for ViT backbone is obtained from one resizing with the solved $d_k^{col}$ and $d_l^{row}$. We directly apply the same set of hyper-parameters chosen based on the OSTrack baseline to another TransT baseline for further validation, and surprisingly find that significant improvements over TransT can be also achieved thanks to the robustness of our method. As acknowledged by Reviewer `Gq4E`, using the same hyper-parameters for all datasets and both trackers is a strength of our paper. Note that the improvements over SwinTrack are also achieved using the same hyper-parameters. We do **respectfully** disagree with the statement that our performance gains seem limited due to the reason that the results in Tab. 3 show some sensitivity to the small set of hyper-parameters in our formulation. Because these hyper-parameters are not independent of or orthogonal to our whole framework. We claim that our formulation also needs to be appropriately set to unveil the power of non-uniform resizing. We will detail the above analysis in the revision.
>
> **About the reasons for using the Gaussian importance scores and the quadratic formulation**: We actually tested another alternative formulation for generating the importance scores before, i.e., generating from the current search image using a saliency detection network and then exploiting the FOVEA [22] grid generator. However, this formulation is time-consuming and we find it difficult to jointly optimize the saliency detection network and the tracking network. We choose the simple yet effective Gaussian importance scores because this formulation is widely used for modeling uncertainty in modern computer vision tasks. As for the quadratic formulation, we need the quadratic energy functions to efficiently solve the non-uniform grid in a Quadratic Programming (QP) formulation (acknowledged by Reviewer `Gq4E`). We will make a clearer explanation in the revision.
>
> **About replacing the non-uniform grid formulation with a simple learning-based model**: Thanks for this insightful comment. Following your suggestion, we tested the case of replacing the QP grid generator with a 3-layer MLP to train the trackers, where the input to the MLP is a flattened 1-D vector from our generated Gaussian importance map. However, this simple learning-based model is indeed hardly able to achieve similar effects to our resizing, obtaining 67.9 AUC on LaSOT. We believe a carefully designed learning-based model and exhaustive experiments are needed to find an effective framework, which can be our future work. We believe our current work can provide good grid generation guidance for this future work.

---

> ### Comment · Reviewer_P4Ao · 2023-08-16
> **Post-rebuttal comments**
>
> With additional experimental evidence to validate their contributions, and further technical details provided by the authors, I consider my issues are adequately addressed by the authors. Thus, I am inclined to raise my rating to weak accept based on the strengths of the paper.

---

> > ### Author Response · Authors · 2023-08-17
> > **Thank Reviewer P4Ao for the reply**
> >
> > We are grateful for your willing to raise the rating to Weak Accept. We will incorporate the additional results and technical details following your suggestions and acknowledge the anonymous reviewers in the revised version of our paper.

---

### Official Review · Reviewer_Fe6p · 2023-07-05

**Soundness:** 4 excellent
**Presentation:** 4 excellent
**Contribution:** 3 good
**Rating:** 8
**Confidence:** 5

**Summary:**

This paper propose a novel method to achieve high tracking speed with smaller input size. Non-uniformly resizing operation is introduced with respect to the object appearing probability.  This resizing operation is then modeled as QP problem and can be effectively soved. This proposed method can be a flexible plugin for multiple deep trackers. Extensive experiments on various tracking benchmarrk are conducted with TransT and OSTrack to demonstrate the effectiveness of the proposed method.

**Strengths:**

1. authors identify the conflict of context ratio and search region size in deep trackers, and proposed a novel method to solve it.

2. the target region resizing is modeled as a QP problem analytically.

3. Introducing non-uniform resizing for target regions and solve the QP problem efficiently.

**Weaknesses:**

1. The tracking performance of proposed non-uniform resizing is only evaluated on speed-oriented tracker. I recommend the author  better to further validate this method on performance-oriented trackers.

2. The computation overhead of proposed method is not reported detailly. A more detail evaluation about the latency of solving the QP problem and resizing image regions with larger context regions should be provided, in term of FLOPS.

3. the effect of proposed non-uniform resizing is not abalated on template and search regions separated.  I think it neccessary to find its effects on template and search regions respectively.

**Questions:**

plz refer to the weakness

**Limitations:**

more ablation studies shoud be conducted.

---

> ### Author Rebuttal · Authors · 2023-08-09
>
> Thank you for your review and feedback! We are happy that you find our method is novel and can be a flexible plugin for other deep trackers. We provide responses to the specific points below:
>
> **About more experimental validations on performance-oriented trackers to further show the generalization capability**: We further apply our method to both the performance-oriented version (backbone: Swin Transformer-Base) and speed-oriented version (backbone: Swin Transformer-Tiny) of SwinTrack [15] to experimentally show the generalization capability of our proposed resizing method for wide range of visual tracking algorithms. Due to the limited computing resources, we only report the comparisons on the large-scale LaSOT dataset (see below), which further clearly show the superiority of our approach. Note that we use the v1 version of SwinTrack as the baseline because only the code of v1 version is publicly available. We will include this new baseline in the revision.
>
> |                          | Size | AUC  | P    |
> | ------------------------ | ---- | ---- | ---- |
> | **Speed-oriented**       |      |      |      |
> | SwinTrack-T-Zoom         | 224  | **68.5** | **72.9** |
> | SwinTrack-T              | 224  | 66.7 | 70.6 |
> | **Performance-oriented** |      |      |      |
> | SwinTrack-B-Zoom         | 224  | **70.5** | **75.4** |
> | SwinTrack-B              | 224  | 69.6 | 74.1 |
> | SwinTrack-B-384          | 384  | 70.2 | 75.3 |
>
> **About detailed evaluation on the latency of solving the QP problem and resizing images with larger context regions**: Thanks for this thoughtful advice. We provide the latency analysis in terms of time (ms) and FLOPs for solving QP and resizing images as follows. It can be seen that resizing images with interpolations in our method costs about half of the total time in spite of its parallel processing, while iteratively solving QP costs the rest of the total time despite its much lower FLOPs. We will include this new analysis in the revision.
>
> |            | Time (ms) | FLOPs |
> | ---------- | --------- | ----- |
> | Solving QP | 0.89      | 0.33M |
> | Resizing   | 0.69      | 2.23M |
> | Total      | 1.58      | 2.56M |
>
> **About investigating the effects of applying the non-uniform resizing on the template and search regions separately**: We are sorry for not clearly indicating that we did not apply our non-uniform resizing on the template image. The template image can be non-uniformly resized by solving QP with the ground truth annotation and a relatively small context factor only once at the beginning frame during testing. Following your suggestion, we have investigated the effects of applying the non-uniform resizing on template and search images separately on LaSOT, and experimentally find that either applying non-uniform resizing on template image alone or on both template and search images performs worse than applying it only on search image (see below). We think the reason is that the template image must have a relatively small context factor to contain minimal necessary context, which makes the whole template region important for the tracking network to locate the target. So the trackers cannot benefit from non-uniform resizing of the template image. We will include this new analysis in the revision.
>
> |             | Zoom     | Zoom   | LaSOT | LaSOT |
> | ----------- | -------- | ------ | ----- | ----- |
> |             | Template | Search | AUC   | P     |
> | OSTrack-256 | √        |        | 64.8  | 71.0  |
> | OSTrack-256 |          | √      | **70.2**  | **76.2**  |
> | OSTrack-256 | √        | √      | 69.6  | 75.5  |

---

> > ### Author Response · Authors · 2023-08-21
> > **Thank Reviewer Fe6p for the increased rating**
> >
> > We are grateful for your increased rating to Strong Accept. We will incorporate the additional results and analyses following your suggestions and acknowledge the anonymous reviewers in the revised version of our paper.

---

### Official Review · Reviewer_PYzX · 2023-07-06

**Soundness:** 3 good
**Presentation:** 3 good
**Contribution:** 2 fair
**Rating:** 5
**Confidence:** 4

**Summary:**

The paper presents a ZoomTrack, a target-aware non-uniform resizing technique for efficient visual tracking. It narrows the gap between speed-oriented and performance-oriented trackers by non-uniformly resizing the cropped image to have a smaller input size while retaining more raw information for the target. A quadratic programming-based formulation for efficient solving is also designed. The experiments demonstrate the consistent improvements over existing transformer trackers.

**Strengths:**

It seems the first time to apply the non-uniform resizing to the field of object tracking.

The paper is well written. The authors provide a clear explanation of the proposed method, including the formulation and solution of the resizing problem through quadratic programming.

The results show that it achieves consistent improvements over two Transformer trackers on multiple tracking datasets.

**Weaknesses:**

The idea is similar to "Learning to Zoom and Unzoom" presented at CVPR 2023.

There are many hyperparameters in the proposed method, and how to set or tune their values?

Although the consistent improvements over two Transformer trackers on multiple tracking datasets are achieved, the improvements on LaSOT and TrackingNet are small.

**Questions:**

See above weaknesses.

**Limitations:**

See above weaknesses.

---

> ### Author Rebuttal · Authors · 2023-08-09
>
> Thank you for your review and feedback! We are happy that you find our paper well written and acknowledge that we are the first time to apply non-uniform resizing to the field of object tracking. We provide responses to the specific points below:
>
> **About the comparison with CVPR 2023 work "Learning to Zoom and Unzoom"**: Thanks for pointing out this concurrent nice work which was presented at CVPR 2023. We find that both this CVPR 2023 work (dubbed as LZU) and our work share a similar idea to improve perception accuracy by non-uniform resizing. However, LZU and our work contribute to different aspects of non-uniform resizing. LZU proposed an efficient and differentiable warp inversion that allows for mapping labels to the warped image to enable the end-to-end training of dense prediction tasks like semantic segmentation. LZU still uses the grid generator from FOVEA [22]. In contrast, our work proposed and carefully designed a new grid generation method, which can improve the quality of non-uniform resizing and perform better than FOVEA in visual tracking. Our method proves to be able to bridge or even close the gap between performance-oriented and speed-oriented trackers. We will cite this concurrent work and include the above analysis in the revised related work.
>
> **About how to set and tune the hyper-parameters in our method**: For the small set of hyper-parameters (acknowledged by Reviewer `Gq4E`) in our QP formulation based on grid representation, we claim that they (e.g., $\beta$, $\gamma$, $\lambda$, grid size) are tightly coupled with our formulation and also have very clear meanings in our framework, so that we can choose a set of reasonable initial values (which may be the optimal values) and then conduct small (coordinate-wise) sweep (acknowledged by Reviewer `Gq4E`) for them to test the sensitivity of our OSTrack-Zoom to these hyper-parameter choices in Tab. 3. For example, by design, we make the zoom factor $\gamma$ capable of controlling the amount of magnification. The performance-oriented version of OSTrack uses 1.5$\times$ larger search-image size than its speed-oriented one, so it is natural to set the initial value of $\gamma$ to 1.5 in our OSTrack-Zoom, which proved to be an appropriate value in our experiments. Another example is the control grid size. It is natural to use 16$\times$16 grid because the search image after non-uniform resizing is evenly split into 16$\times$16 patches and then sent into the Vision Transformer (ViT) backbone for feature extraction. As our resizing process can be intuitively understood as resizing each rectangular grid patch on the source image to the one on the target image, we can expect that each patch for ViT backbone is obtained from one resizing with the solved $d_k^{col}$ and $d_l^{row}$. We directly apply the same set of hyper-parameters chosen based on the OSTrack baseline to another TransT baseline for further validation, and surprisingly find that significant improvements over TransT can be also achieved thanks to the robustness of our method. As acknowledged by Reviewer `Gq4E`, using the same hyper-parameters for all datasets and both trackers is a strength of our paper. Note that the improvements over SwinTrack (see Tab. R3) are also achieved using the same hyper-parameters. We will detail the above analysis in the revision.
>
> **About the performance gains on LaSOT and TrackingNet**: Although our performance gains on TrackingNet are small, we think that the AUC improvements on LaSOT with respect to both baselines (1.1 for OSTrack and 2.2 for TransT) are still impressive. As acknowledged by Reveiwer `2uk1`, enabling OSTrack to achieve 70.2 AUC on LaSOT with 100fps is impressive. As for the reason of small improvements on TrackingNet, we additionally analyze the ratios of challenging scenarios appearing in the different tracking datasets (see below), which clearly show that the test split of TrackingNet has a very different property from LaSOT, LaSOT_ext, and TNL2K with respect to the ratios of challenging scenarios. As shown in Fig. S1 of our appendix material, our method can achieve significant performance gains under challenging scenarios with drastic movement (see Fast Motion (FM), Viewpoint Change (VC)) or significant drift after some period of time (see Full Occlusion (FOC), Out-of-View (OV)) owning to the enlarged visual field. However, the ratios of the challenging scenarios FOC, OV, FM in TrackingNet (VC is not labeled in TrackingNet) are significantly lower than in LaSOT, LaSOT_ext, and TNL2K, which may be the reason for small improvements of our approach on TrackingNet. We will include this new analysis in the revision.
>
> | Dataset     | Ratio of FOC | Ratio of OV | Ratio of FM | Ratio of VC     | AUC Gain (OSTrack) |
> | ----------- | ------------ | ----------- | ----------- | --------------- | ------------------ |
> | LaSOT       | 42.1%        | 37.1%       | 18.9%       | 11.8%           | +1.1%              |
> | LaSOT_ext   | 62.7%        | 21.3%       | 58.7%       | 39.3%           | +3.1%              |
> | TNL2K       | 13.9%        | 33.4%       | 24.0%         | 44.3%           | +2.2%              |
> | TrackingNet | 4.7%         | 4.0%          | 10.0%         | - (not labeled) | +0.1%              |

---

### Official Review · Reviewer_Gq4E · 2023-07-11

**Soundness:** 3 good
**Presentation:** 4 excellent
**Contribution:** 2 fair
**Rating:** 5
**Confidence:** 4

**Summary:**

This paper proposes a simple but clever method to improve the accuracy-speed trade-off of state-of-the-art methods for single object tracking. Existing methods can achieve greater accuracy by using higher-resolution images, however this comes at the cost of inference speed. This paper proposes to adopt a non-uniform resizing of the image which preserves a large search region without sacrificing the resolution of the image at the location where the object is expected to appear. Solving for the control points of an axis-aligned grid is formulated as a QP to preserve the bounds of the region while minimizing distortion and approximating a desired zoom in the region where the object is likely to appear. Results on several large datasets with two different trackers show that this simple modification makes a large step towards closing the gap between fast and accurate trackers. The small set of hyper-parameters is fixed for all datasets and both trackers.

**Strengths:**

1. Paper identifies a practical dilemma and proposes a simple but clever solution.
1. Proposed method can be applied with many different trackers.
1. The method significantly advances the Pareto front of the AUC vs MACs trade-off, even dominating the best trackers on the TNL2K dataset (Figure 1).
1. Solving for a non-uniform grid is formulated as a QP with efficient solution.
1. Small (coordinate-wise) sweep conducted for the key hyper-parameters on large dataset.
1. Same hyper-parameters used for multiple datasets and multiple methods.
1. The visualizations show that the target is magnified without being distorted.
1. The statistics demonstrate that, under the proposed non-uniform resizing, the mean size of the target is increased while its variance is unaffected.
1. Inclusion of FOVEA as baseline.
1. Increase in time required by non-uniform resize operation was accounted for.

**Weaknesses:**

1. My main concern is that the proposed non-uniform resizing may be similar in effect to varying the crop hyper-parameters (method for setting $c_x$ and $c_y$, search-image context factor, template-image size) and there is not a rigorous comparison. Showing that the proposed method (optimized for $\beta$, $\gamma$, $\lambda$) is better than optimizing these hyper-parameters would be much more convincing.
1. No confidence intervals (error bars). This would be particularly useful for Table 3.
1. Figure 3 shows that increasing the search-image context factor results in worse accuracy due to the low resolution of the object. However, the drop in accuracy may also be caused by a mismatch in the size of the object in the search and template images. The same applies for decreasing the search-image context factor. I think this study should also consider the template-image size.
1. The paper did not explain how the non-uniform resizing was applied to the search and template images (question below).
1. It is concerning that a higher-resolution control grid leads to worse results. This suggests that the QP objective does not capture all necessary factors and instead relies on a coarse grid to avoid over-optimizing the flawed objective.
1. It seems unnatural to take the square root of the box width in $\sigma_x = \sqrt{\beta \cdot r^w}$. This would result in $\sigma$ depending non-linearly on the box size. Is there a reason for this?
1. Code for experiments not provided.

**Questions:**

Please provide counter-arguments to (or correct misunderstandings in) the weaknesses above.

1. How is the non-uniform resizing applied to the search and template images? Was it applied to the template image? If so, is the equation solved again with different W and H? Could this result in the object having a different size in the search and template image?
1. Does FOVEA have a hyper-parameter that would improve its results (or could it be trivially modified to include such a hyper-parameter)? Would the proposed warping still outperform it?
1. In the experiments, it is stated that the context factor for the search image was increased to 5. This was shown to be optimal for the baseline OSTrack on LaSOT in Figure 3 (~69.5 AUC compared to ~69.0 AUC at 4.5). Table 1 shows OSTrack-256 to achieve 69.1 AUC on LaSOT. This suggests that the optimal context factor was not used for the baseline. Is this because a context factor of 5 has a negative effect on the baseline on other datasets? (OSTrack-Zoom achieved 70.2 AUC, which is less impressive compared to 69.5 than compared to 69.0.)

Less important:
1. What is the effect of applying ZoomTrack to the performance-oriented trackers like OSTrack-384?
1. I'm curious about how important it is to re-train the tracker. What is the effect of applying the non-uniform resizing _without_ re-training the tracker?

Suggestion:
1. If the object was treated as a square, then the QP could be solved once offline for all objects. Would this have a significant negative impact on the method?
1. Is the solution of the QP always symmetric in x and y?

Edits:
1. The term "performance" is multivalent (may mean speed or accuracy); I suggest "accuracy" or "quality" instead.
1. "throw the question" -> "pose the question"
1. "with possibility" -> "with probability"
1. Citation required for 100MB / 1MB in human vision system.
1. It is stated that the context factor for the template image is left unchanged while that for the search image is increased to 5. Please state the original values so that the reader does not need to go and find them.

**Limitations:**

The authors have adequately addressed the limitations of the work. I do not foresee any negative impacts stemming from this specific work.

---

> ### Author Rebuttal · Authors · 2023-08-09
>
> Thank you for your review and feedback! *We will release our code upon acceptance and include the new analyses below in the revision*. We provide responses to the specific points below:
>
> **About the edits**: We will revise the edits accordingly. The citation for 100MB/1MB in the human vision system is [29] in the main paper. The original context factors for the template and search images are 2 and 4 respectively.
>
> **About our superiority over the uniform resizing baseline with its crop hyper-parameters optimized**: Following your suggestion, we additionally explore whether optimizing the crop hyper-parameters in the uniform resizing baseline (See Tab. R1 in the attached PDF for “global” response) can have a similar effect to our proposed non-uniform resizing. Despite the fact that our limited compute resources make us unable to optimize the above hyper-parameters on large tracking datasets in a grid-search manner, our adequately powered experiments clearly show that the above optimizing is indeed hardly able to achieve similar effects to our resizing. Even for the baseline OSTrack-256 with the optimal crop hyper-parameters on LaSOT (69.5 AUC), we experimentally find that it can only achieve AUC gains of +0.4 and +0.6 on LaSOT_ext and TNL2K respectively while performing worse than the original OSTrack-256 on TrackingNet (83.1 SUC vs 82.1 SUC). In other words, our method still outperforms the above optimal baseline on LaSOT (+0.7 AUC), LaSOT_ext (+2.7 AUC), TNL2K (+1.6 AUC), TrackingNet (+1.1 SUC). Moreover, it is experimentally shown that the mismatch of the object size between the search and template images caused by increasing the search-image context factor has a relatively small effect on the final tracking accuracy for modern transformer trackers. That means the low resolution of the object is the major cause for the worse accuracy.
>
> **About how ZoomTrack was applied to the search and template images? Can QP be solved once offline for all objects?**: We are sorry for not clearly indicating that we did not apply our non-uniform resizing on the template image. Although the template image can be non-uniformly resized by solving QP with the ground truth annotation and a relatively small context factor only once at the beginning frame during testing, we experimentally find that either applying non-uniform resizing on template image alone or on both template and search images performs worse than applying it only on search image (Tab. R2). We think the reason is that the template image must have a relatively small context factor to contain minimal necessary context, which makes the whole template region important for the tracking network to locate the target. So the trackers cannot benefit from non-uniform resizing of template image. As for the search image, the QP is solved for every frame given the previous prediction box during tracking. Following your suggestion, we tested the case of treating the object as a square so as to solve QP once offline for all objects. However, AUC on LaSOT drops from 70.2 to 69.7. We leave the thorough investigation on this issue for future work due to the limited rebuttal time.
>
> **About the concern of denser grid leading to worse results**: As we mentioned in the main paper (Lines 362-363), we think the reason for the worse results is that the important areas generated from the temporal prior during tracking are not always accurate, which is inevitable since the prediction of the tracker cannot be always correct. We think it is the inaccurate previous frame prediction that leads to performance degradation for the denser grid rather than our QP objective. To validate this argument, we tested the case of using the ground truth from the previous frame instead of the predicted bounding box to crop and resize the search region for the current frame. Under this new setting, our method using grid size 32 outperforms 16 counterpart (77.6 AUC vs 76.9 AUC) on LaSOT, proving that denser grid can achieve better accuracy thanks to the more reliable temporal prior.
>
> **About the error bars for Tab. 3**: Theoretically, neither the deterministic OSTrack nor our resizing module introduces randomness during testing. We also ran our tracker 3 times on LaSOT and the results are the same. As for the training with stochastic gradient descent, we trained the models for all the experiments using a shared random seed. We leave the training using different random seeds for error bars for future work due to the limited rebuttal time.
>
> **About $\sigma_x$ in Eq. (5)**: We take the square root of the box width for $\sigma_x$ following the practice in FOVEA [22] which converts bounding boxes to a saliency map based on kernel density estimation.
>
> **About tuning hyper-parameters for FOVEA**: FOVEA shares some hyper-parameters with our method, e.g., grid size and bandwidth. We tuned them when preparing our work and interestingly found that its best performance on LaSOT (67.6 AUC) is also achieved by setting grid size to 16 and bandwidth to 64, which are the same as our method.
>
> **About applying ZoomTrack to performance-oriented trackers like OSTrack-384**: We tested the case of increasing the context factor of OSTrack-384 from 5 to 6 while increasing its search-image size to 480, however, it cannot further improve AUC on LaSOT (71.1 vs 71.0). Thus it is difficult for ZoomTrack to further improve over OSTrack-384. However, our improvements over SwinTrack [15] on LaSOT are also impressive. Please refer to the 3rd item of the “global” response for more details.
>
> **About the effect of applying ZoomTrack without re-training**: In Sec. D of our supplementary material we have studied this effect. Please refer to this material for more details.
>
> **About the symmetric QP solution**: The solution in the x-axis and y-axis is always symmetric because the Gaussian importance score function for the search image is symmetric in x and y. Some accelerating strategies may be developed using this symmetry.

---

> > ### Comment · Reviewer_Gq4E · 2023-08-21
> >
> > Thank you to the authors for a detailed rebuttal including several valuable experiments.
> >
> > > (insufficiently addressed) Error bars
> >
> > I'm not referring to randomness in the inference algorithm, but to treating the test set as a finite sample from an iid distribution. Confidence intervals could be obtained using the variance of the sample mean (or bootstrap sampling).
> >
> > > Crop hyper-parameters of baseline
> >
> > Thank you very much for conducting this experiment. I trust that the higher score for the baseline with the same parameters (5/256; 69.5 AUC) will be included in the final version. While the gap for LaSOT is smaller, the method still achieves a consistent improvement across all datasets compared to the improved baseline.
> >
> > > Solving QP once for square object
> >
> > Thanks for conducting this experiment!
> >
> > > (insufficiently addressed) Square root of $\sigma_x$
> >
> > I still find this irregular and unsatisfying, as it means the behaviour is not scale-invariant, however it's not a dealbreaker.
> >
> > **Overall**
> >
> > The paper proposes a simple and reliable technique that can be applied to obtain a small but significant improvement for various different trackers. The empirical evaluation is comprehensive, fair and easy to understand. For now, I retain my initial rating of borderline accept.

---

> > > ### Author Response · Authors · 2023-08-21
> > > **Thank Reviewer Gq4E for the reply**
> > >
> > > Thank you for your careful review and thoughtful reply, and we are happy to continue the discussion. We address the two insufficiently addressed issues separately.
> > >
> > > **About the error bars for Tab. 3**: Thank you very much for pointing out our misunderstanding of the way to obtain error bars. Following your suggestions, we calculate the 90% confidence intervals for all the experiments in Tab. 3 using the bootstrap sampling method. Each of the 90% confidence intervals is obtained using 10000 bootstrap samples, each of which has the size of 280 (the same to the sequence number of LaSOT-test-split). The new results with error bars are listed below. We will include these new results in the final version.
> > >
> > > | $\beta$  | 4             | 64            | 256           |
> > > | ----- | ------------- | ------------- | ------------- |
> > > | upper | 70.95 (+1.87) | 72.02 (+1.86) | 71.54 (+1.90) |
> > > | mean  | 69.08         | 70.16         | 69.64         |
> > > | lower | 67.23 (-1.85) | 68.33 (-1.83) | 67.80 (-1.84) |
> > >
> > > | Grid Size  | 8             | 16            | 32            |
> > > | ----- | ------------- | ------------- | ------------- |
> > > | upper | 68.50 (+2.06) | 72.02 (+1.86) | 71.52 (+1.98) |
> > > | mean  | 66.44         | 70.16         | 69.54         |
> > > | lower | 64.49 (-1.95) | 68.33 (-1.83) | 67.63 (-1.91) |
> > >
> > > | $\gamma$ | 1.25          | 1.5           | 1.75          |
> > > | ----- | ------------- | ------------- | ------------- |
> > > | upper | 71.58 (+1.89) | 72.02 (+1.86) | 71.35 (+1.94) |
> > > | mean  | 69.69         | 70.16         | 69.41         |
> > > | lower | 67.88 (-1.81) | 68.33 (-1.83) | 67.49 (-1.92) |
> > >
> > > | $\lambda$ | 0             | 1             | 2             |
> > > | ------ | ------------- | ------------- | ------------- |
> > > | upper  | 71.69 (+1.89) | 72.02 (+1.86) | 71.43 (+1.87) |
> > > | mean   | 69.80         | 70.16         | 69.56         |
> > > | lower  | 67.93 (-1.87) | 68.33 (-1.83) | 67.71 (-1.85) |
> > >
> > > **About the square root of $\sigma_x$**: Although FOVEA [22] did not explain the reason for taking the square root of the temporal prior bounding box width $r^w$ for $\sigma_x$, we are sorry for not providing our own explanation for that. In our opinion, it may be attributed to the reason that the temporal priors extracted from previous frame’s predictions could include some noises and taking the square root may reduce the impacts of these noises thanks to the improved signal-to-noise ratio. For instance, considering the temporal prior bounding box width $r^w=r^w_{gt}+r^w_{noise}$ consisting of the ground-truth value $r^w_{gt}$ and the noise $r^w_{noise}$, if the noise can reach up to ±10% of its ground-truth value, the original $r^w$ will lie in (0.9$r^w_{gt}$\~1.1$r^w_{gt}$) while its square root will lie in (0.95$\sqrt{r^w_{gt}}$\~1.05$\sqrt{r^w_{gt}}$). We think this benefit of reducing impacts of noises may outweigh the missing of scale-invariant behaviour.

---

### Author Rebuttal · Authors · 2023-08-09

Thank you for the valuable reviews pointing out that our *simple but clever* (`Gq4E`) and *novel* (`Fe6p`) method, focusing on *a very interesting problem which is little-explored* (`2uk1`) and *easily overlooked* (`P4Ao`), *makes a large step towards closing the gap between fast and accurate trackers and advancing the Pareto front of the AUC vs MACs trade-off* (`Gq4E`) and is also *the first time* (`PYzX`) applied to the tracking field while *bringing adequate enlightenment* (`P4Ao`), and acknowledging that our *technically plausible* (`P4Ao`), *elegant* (`2uk1`) and *clearly explained* (`PYzX`) solution can achieve *impressive* (`2uk1`) and *consistent improvements* (`PYzX`) over the baselines on most of the tracking datasets. Prompted by the insightful reviews, we mainly present the following additional experimental results and analyses for the common questions:

- Following the suggestion by Reviewer `Gq4E`, we additionally explore whether optimizing the original crop hyper-parameters in the uniform resizing baseline (i.e., setting $c^w$ and $c^h$ using two different choices as in Sec. 3.1, varying the search-image context factor, and varying the template-image size to alleviate the mismatch of the object size between the search and temple images) can have a similar effect to our proposed non-uniform resizing. Despite the fact that our limited compute resources make us unable to optimize the above hyper-parameters on large tracking datasets in a grid-search manner, our adequately powered experiments clearly show that the above optimizing is indeed hardly able to achieve similar effects to our resizing (See Tab. R1 in the attached PDF). Moreover, it is experimentally shown that the mismatch of the object size between the search and temple images caused by increasing the search-image context factor has a relatively small effect on the final tracking accuracy for modern transformer trackers.

- As for the confusion of applying the non-uniform resizing on the cropped template (raised by Reviewers `Gq4E`, `Fe6p` and `2uk1`), we are sorry for not clearly indicating that we did not apply our non-uniform resizing on the template image. The reason is that we experimentally find that either applying non-uniform resizing on the template image alone or on both template and search images performs worse than applying it only on the search image (see Tab. R2). We think the reason is that the template image must have a relatively small context factor to contain minimal necessary context, which makes the whole template region important for the tracking network to locate the target. So the trackers cannot benefit from non-uniform resizing of the template image.

- Following the suggestions by Reviewers `Gq4E`, `Fe6p`, `P4Ao`, and `2uk1`, we further apply our method to both the performance-oriented version (backbone: Swin Transformer-Base) and speed-oriented version (backbone: Swin Transformer-Tiny) of SwinTrack [15] to experimentally show the generalization capability of our proposed resizing method for wide range of visual tracking algorithms. Due to the limited computing resources, we only report the comparisons on the large-scale LaSOT dataset (See Tab. R3), which further clearly show the superiority of our approach. Although analyzing the importance of choosing input resizing method is an overlooked and little-explored problem in the visual tracking field and the common practice is using uniform resizing by default, we believe our further analysis based on SwinTrack, as well as the previously explored two baselines (OSTrack and TransT), can demonstrate the importance of choosing appropriate input resizing method.

- As for the concern of hyper-parameters (e.g., $\beta$, $\gamma$, $\lambda$) in our formulation (raised by Reviewers `PYzX` and `P4Ao`), we claim that this small set of hyper-parameters (acknowledged by Reviewer `Gq4E`) is tightly coupled with our formulation and also has very clear meanings, so that we can choose a set of reasonable initial values (which may be the optimal values) and then conduct small (coordinate-wise) sweep (acknowledged by Reviewer `Gq4E`) for them to test the sensitivity of our OSTrack-Zoom to these hyper-parameter choices in Tab. 3. Please refer to the separate responses for more details. We then directly apply our method using the same set of hyper-parameters on TransT and find significant improvements thanks to the robustness of our method. As acknowledged by Reviewer `Gq4E`, using the same hyper-parameters for all datasets and both trackers is a strength of our paper. Note that the improvements over SwinTrack (see Tab. R3) are also achieved using the same hyper-parameters. We do **respectfully** disagree with Reviewer `P4Ao` on the statement that our performance gains seem limited for the reason that the results in Tab. 3 show some sensitivity to the small set of hyper-parameters in our formulation. Because these hyper-parameters are not independent of or orthogonal to our whole framework. We claim that our formulation also needs to be appropriately set to unveil the power of non-uniform resizing.

- As for the concern of small improvements on TrackingNet (raised by Reviewers `PYzX` and `2uk1`), we additionally analyze the ratios of challenging scenarios appearing in the different tracking datasets (See Tab. R4), which clearly show that the test split of TrackingNet has a very different property from LaSOT, LaSOT_ext, and TNL2K with respect to the ratios of challenging scenarios. As shown in Fig. S1 of our appendix, our method can achieve significant performance gains under challenging scenarios such as Fast Motion (FM), Viewpoint Change (VC), Full Occlusion (FOC), Out-of-View (OV), owning to the enlarged visual field. However, the ratios of FOC, OV, FM in TrackingNet (VC is not labeled in TrackingNet) are significantly lower than in LaSOT, LaSOT_ext, and TNL2K, which may be the reason for small improvements of our approach on TrackingNet.

---

### Decision · Program_Chairs · 2023-09-21

**Decision:**

Accept (spotlight)

**Comment:**

The paper proposes an elegant non-uniform resizing method to retain more raw information for the target area when resizing images with large contexts. By controlling the magnification of important areas and restricting extreme deformation, the resizing is efficiently achieved via quadratic programming. Extensive experiments on challenging benchmarks demonstrate consistent and significant improvements in the accuracy-speed trade-off over strong baselines including both CNN and transformer trackers. The reviewers positively acknowledge the novelty, soundness, clarity, and impact of this technically solid work. ACs concur with the reviewers' assessments and recommend accepting this paper.